# DFPC: Data Flow driven Pruning of Coupled channels without data.

**Tanay Narshana**[1*]**, Chaitanya Murti**[2]**, and Chiranjib Bhattacharyya**[2,3]
Observe.AI[1]
Robert Bosch Centre for Cyber-Physical Systems, Indian Institute of Science[2]
Department of Computer Science and Automation, Indian Institute of Science[3]
`tanay.narshana@gmail.com`, `{mchaitanya, chiru}@iisc.ac.in`

## Abstract

Modern, *multi-branched* neural network architectures often possess complex interconnections between layers, which we call *coupled channels* (CCs). Structured pruning of CCs in these multi-branch networks is an under-researched problem, as most existing works are typically designed for pruning single-branch models like VGG-nets. While these methods yield accurate subnetworks, the improvements in inference times when applied to multi-branch networks are comparatively modest, as these methods do not prune CCs, which we observe contribute significantly to inference time. For instance, layers with CCs as input or output take more than 66% of the inference time in ResNet-50. Moreover, pruning in the data-free regime, where data is not used for pruning, is gaining traction owing to privacy concerns and computational costs associated with fine-tuning. Motivated by this, we study the problem of pruning CCs in the data-free regime. To facilitate the development of algorithms to prune CCs, we define Data Flow Couplings (DFCs) to enumerate the layers that constitute coupled connections and the associated transformation. Additionally, saliencies for pruning CCs cannot be gauged in isolation, as there may be discrepancies among the layerwise importance of CCs using conventional scoring strategies. This necessitates finding *grouped saliencies* to gauge the importance of all corresponding coupled elements in a network. We thus propose the Backwards Graph-based Saliency Computation (BGSC) algorithm, a data-free method that computes saliencies by estimating an upper bound to the reconstruction error of intermediate layers; we call this pruning strategy *Data Flow driven Pruning of Coupled channels (DFPC)*. Finally, we show the efficacy of DFPC for models trained on standard datasets. Since we pruned coupled channels, we achieve up to 1.66x improvements in inference time for ResNet-101 trained on CIFAR-10 with a 5% accuracy drop without fine-tuning. With access to the ImageNet training set, we achieve significant improvements over the data-free method and see an improvement of at least 47.1% in speedup for a 2.3% accuracy drop for ResNet-50 against our baselines.[1]

## 1 Introduction

As computational resources have become significantly more powerful, deep learning models have become correspondingly larger and more complex as well, with some models possessing billions of parameters (Sevilla et al., 2022). Moreover, many modern architectures are *multi-branched* networks due to layer skip connections like Residual Connections (He et al., 2016) that are used to avoid vanishing gradients. These large, complex architectures enable these models to learn patterns in data with better performance in terms of optimization and generalization (Arora et al., 2019; Neyshabur et al., 2019; Zhang et al., 2021).

The benefits of overparameterization in these models come at the cost of increased memory and compute footprint, necessitating the invention of techniques to mitigate them. Techniques such as

---

*Work done at the Department of Computer Science and Automation, Indian Institute of Science.
[1]Our code is publicly available at https://github.com/TanayNarshana/DFPC-Pruning

network pruning(Hoefler et al., 2021), quantization(Gholami et al., 2021), knowledge distillation(Gou et al., 2021), and low-rank decomposition(Jaderberg et al., 2014) make it possible to compress overparameterized models in order to improve real-world performance metrics such as inference time and power consumption. Pruning involves discarding elements of neural networks after gauging the importance or saliencies of these elements. Generally, two broad categories of pruning techniques exist in the literature - unstructured pruning, which involves removing individual weights from the model, such as the results in Han et al. (2015); LeCun et al. (1989); Tanaka et al. (2020), and structured pruning (also called channel pruning for CNNs), which involves removing entire neurons or channels from the model (Ding et al., 2021; Luo et al., 2017; Prakash et al., 2019; Singh et al., 2019; Wang et al., 2021; He et al., 2017). In this work, we focus on structured pruning for multi-branched CNNs.

Due to the complicated interconnections that exist in multi-branched networks, pruning multi-branched neural networks such as ResNets and MobileNets, raise unique challenges that do not arise when pruning *single branch* networks such as VGG-nets (Simonyan & Zisserman, 2015). These complex connections, such as residual connections in ResNets, require channels fed into the connection to be of the same dimensions, thus *coupling the channels*. Pruning such coupled channels (CCs) is generally not addressed in current works on structured pruning, such as Ding et al. (2021); Joo et al. (2021); Luo et al. (2017); Singh et al. (2019); Wang et al. (2021), which are designed for pruning single-branched networks; for example, in ResNets, only the output channels of the first two layers of a ResNet residual block are pruned, and the channels that feed into the residual connections are ignored when using these methods. Pruning CCs is challenging since not pruning filters from all the associated layers would break the CNN. Furthermore, pruning CCs is crucial as we observe that the layers associated with CCs take up a significant portion of the inference time - more than 66% in ResNet-50.

The few methods for pruning CCs currently available generally rely on data-driven statistics of the output layer to infer saliencies and involve heavy finetuning(Chen et al., 2021; Liu et al., 2021; Luo & Wu, 2020; Shen et al., 2021). However, situations may arise where models trained on proprietary datasets may be distributed but not the dataset for reasons such as privacy, security, and competitive disadvantage(Yin et al., 2020). Thus, pruning without data is an important challenge and an active area of research(Patil & Dovrolis, 2021; Srinivas & Babu, 2015; Tanaka et al., 2020). However, these techniques do not address pruning CCs, especially in the one-shot and data-free pruning regime, which is an open problem(Hoefler et al., 2021).

In this work, we aim to prune CCs with the additional challenge of doing so without access to data. Towards answering the posed challenges, our **contributions** in this work are as follows.

1. Unlike single-branch networks, the CCs in multi-branched networks provide an additional challenge for structured pruning. Identifying the associations between coupled layers, as well as the mappings between them, is a nontrivial task. To address this problem, we define *Data Flow Couplings (DFCs)* to abstract the notion of coupling in a network by enumerating both the layers and transformations involved in the coupling. Two types of layers are associated with a DFC - feed-in and feed-out layers; their outputs and inputs are involved in the coupling.

2. Pruning involves measuring saliencies for elements to be pruned, which, for CCs, is not straightforward due to interconnections between layers. In Section 4, we investigate whether saliencies of channels in a DFC can be inferred in isolation from the feed-in layers. To do so, we define *Maximum Score Disagreement* to quantify disagreement in saliencies among the feed-in layers. We empirically observe significant disagreement among saliencies assigned to channels by the feed-in layers suggesting that the importance of such channels cannot be deduced in isolation. This leads us to propose *grouped saliencies*, with which we can rank coupled elements of a DFC.

3. Measuring the effect of pruning coupled elements of a multi-branch network without data, and thus inferring filter saliencies, is a challenging task. For this, Theorem 1 proposes a saliency mechanism, using the transformations enumerated by the DFCs, that bounds the joint reconstruction error of the outputs of the DFC's feed-out layers without data. To compute these saliencies, we propose the Backwards Graph-based Saliency Computation (BGSC)[1] Algorithm. To mitigate the computational cost of this algorithm (both time and memory) for CNNs, we provide a parallelized implementation of the algorithm owing to its *embarrassingly parallel* nature. On ResNet-101 for the CIFAR-10 dataset, we obtain a **1.66x** inference time speedup for a **5%** accuracy drop without

 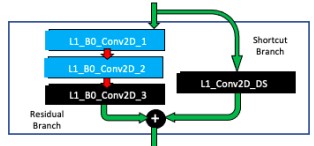 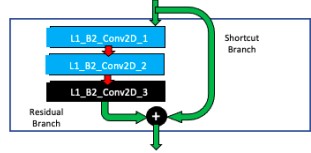

(a) A feed-forward neural network     (b) Residual block in ResNet-50     (c) Residual block in ResNet-50 with
                                                      with downsample layer                  no layer in shortcut branch

Figure 1: Examples of data flow (sub)graphs in various networks

retraining. On ResNet-50 for the ImageNet dataset, we obtain an inference time speedup of at least **47.1%** against our baselines for a **2.3%** accuracy drop with retraining.

## 2    NOTATION, PRELIMINARIES AND RELATED WORK

### 2.1    NOTATION AND PRELIMINARIES

**Notation**. Let $[n] = \{1, ..., n\} \subset \mathbb{N}$ for any $n \in \mathbb{N}$. Let $\mathbf{x} \in \mathbb{R}^n$ denote a vector, and $M$ denote a matrix. $x(k)$ and $M(i, j)$ denote the $k^{th}$ and $(i, j)^{th}$ element of $\mathbf{x}$ and $M$ respectively. $\|\mathbf{x}\|_1$ and $\|\mathbf{x}\|_\infty$ denote the L1-norm and max-norm of the vector $\mathbf{x}$ respectively. $|\mathbf{x}|/|M|$ denote the element-wise absolute (vector of $\mathbf{x}$)/(matrix of $M$). That is, $|x|(k) = |x(k)|$ and $|M|(i, j) = |M(i, j)|$. $\odot$ denotes the Hadamard product. The output of an element-wise transformation $F$ applied on a matrix is such that the $(i, j)^{th}$ element of the output depends only on the $(i, j)^{th}$ element of the input. That is, for matrix $M$, $F(M)(i, j) = f_{(i,j)}(M(i, j))$. A similar definition is used in this manuscript when considering element-wise transformations on a vector and when the element-wise transformation is a function of multiple matrices/vectors. Let $|A|$ denote the cardinality of a set $A$.

**Networks under consideration for analysis**. Across definitions and derivations in Sections 3 and 5, we assume that fully-connected (FC) layers are the only layers in the network that do not perform element-wise transformations. The network can have layers like batch-norm and non-linearities like ReLU that perform element-wise transformations. Let there be $L$ FC layers in the network. We assign each FC layer a unique integer from the set $[L]$. The order of numbering is not important. Consider a layer $l \in [L]$. We denote its weight matrix by $\mathbf{W}_l$. For a layer $l$ that is given an input $\mathbf{x}$, the corresponding output $\mathbf{y}$ is obtained as $\mathbf{y} = \mathbf{W}_l\mathbf{x}$.

**Terminology for CNNs**. We recall the standard definitions from Hoefler et al. (2021); Dumoulin & Visin (2016). *Channels* denote the different views of the data (e.g., the red, green and blue channels of a color image). Convolutional layers have multiple *filters*, each comprising multiple *kernels*. A convolutional layer with $n$ input and $m$ output channels has $m$ filters, each comprising $n$ kernels.

**Data flow graph**. The *data flow graph* of a neural network is a directed graph that encapsulates the transformation produced by a network. Each node in the graph applies some operation to the data provided. Each edge in the graph denotes the data flow between nodes, with data flowing from tail to head of the edge. *The backwards graph of the neural network is similar to the data flow graph except for the direction of edges being reversed.* Figure 1a shows the data flow graph of a four-layer feed-forward neural network. Such a network is said to have a single branch.

**Terminology for ResNets**. ResNets consist of Residual Connections(He et al., 2016). A block of layers in these networks with a residual connection around them is called a *residual block*. Figures 1b and 1c show instances of residual blocks in ResNet-50. These have two branches and thus make ResNets multi-branched. The *residual branch* contains most of the convolutional layers in the residual block. The other branch is called the *shortcut branch*. In ResNets, multiple consecutive residual blocks are clubbed together and called a *layer-block*.

### 2.2    RELATED WORK

**Pruning coupled channels (CCs).** Existing literature (Gao et al., 2019; Liu et al., 2021; Luo & Wu, 2020; Shen et al., 2021) prune CCs by grouping layers whose output channels are coupled. Liu et al. (2021) propose an algorithm to group such layers. Liu et al. (2021); Luo & Wu (2020); Shen et al. (2021) utilize data-driven statistics of the output layer to measure saliencies. Gao

et al. (2019), and Chen et al. (2021) alter the training objective by introducing sparsity-promoting regularizers. But, Gao et al. (2019) use the Train-Prune-Finetune pruning pipeline(Han et al., 2015) whereas Chen et al. (2021) simultaneously train and prune the network. The experimental results, particularly of Liu et al. (2021); Shen et al. (2021), show that pruning CCs results in a better trade-off between accuracy and inference time speedups when finetuning is possible.

**Saliency scores.** Techniques exist in structured pruning that derive saliencies of channels from the information extracted from consecutive layers(Joo et al., 2021; Luo et al., 2017) or one layer(Hu et al., 2016; Li et al., 2017) without access to a dataset. Such structured pruning techniques locally measure the saliencies of channels. Gao et al. (2019) and Yang et al. (2018) utilize joint norms of weights in filters of grouped layers to infer saliencies. Minimizing the Reconstruction Error of the next layer is a metric to gauge the saliencies of channels(Luo et al., 2017; Yu et al., 2018) in structured pruning. However, this metric has only been applied to prune non-CCs and is assumed in the literature to not apply to CCs since such a metric requires a notion of consecutiveness among layers(Liu et al., 2021). However, in this work, we leverage DFCs to solve this problem.

**Data-free pruning.** Early efforts toward data-free pruning include Srinivas & Babu (2015), which measured similarity between neurons and merged similar neurons. Recently, Tanaka et al. (2020) proposed the Synflow method, an unstructured, data-free pruning method that relied on preserving gradient flow. Similar works include Gebhart et al. (2021); Patil & Dovrolis (2021), which use the Neural Tangent Kernel-based techniques to modify SynFlow. Yin et al. (2020) synthesize a dataset from a pre-trained CNN classifier and utilize the synthesized dataset to perform iterative data-driven pruning, with the drawback that synthesizing the dataset from the classifier is very costly.

## 3 DATA FLOW COUPLINGS

In this section, motivated by the need to enumerate both the layers involved in CCs as well as the transformations between them, we define Data Flow Couplings (DFCs). We also provide examples of DFCs in ResNet-50.

**Motivation for defining DFCs**. Studies intending to prune CCs in CNNs either group layers whose output channels are coupled(Gao et al., 2019; Liu et al., 2021; Luo & Wu, 2020; Shen et al., 2021) or group weights across layers if they belong to the same coupled channel(Chen et al., 2021). However, these groupings do not simultaneously enumerate both the layers and the transformations involved in an instance of coupling. Such descriptions aid us in understanding the end-to-end transformation produced by the instance of coupling. In Section 5, we use these descriptions to derive a data-free mechanism for which DFCs are crucial. A *Data Flow Coupling (DFC)* abstracts the end-to-end transformation and the associated layers for an instance of coupling in a network.

**Definition 1** *Consider a neural network with $L$ FC layers where each FC layer is assigned a unique integer from the set $[L]$. Consider two sets of layers $A = \{a_1, a_2, ..., a_p\}, B = \{b_1, b_2, ..., b_q\}$ where $A, B \subset [L]$. Let $\mathbf{z}^{(m)}$ be an arbitrary input sample from a data set $\{\mathbf{z}^j\}_{j=1}^M$ that is fed to the network, let $\mathbf{u}_a^{(m)}$, $\mathbf{v}_a^{(m)}$ denote the input to and the corresponding output of layer $a \in A$, and let $\mathbf{x}_b^{(m)}$, $\mathbf{y}_b^{(m)}$ denote the same for layer $b \in B$. Suppose there exists a collection of functions $F$ defined by the data flow graph of the network, such that the input to any layer $b \in B$ is obtained through a map $\mathbf{F}_b : \mathbb{R}^{\sum_{a \in A} dim(\mathbf{v}_a^{(m)})} \to \mathbb{R}^{dim(\mathbf{x}_b^{(m)})} \in F$, where $\mathbf{F}_b$ is a function of the outputs of layers $a \in A$. Let the function that gives the value of activation to the $k^{th}$ neuron in $\mathbf{F}_b$ be denoted by $F_{bk}$. Then, we say the tuple $\tau = <A, B, F>$ is a **data flow coupling** if*

($C_1$) $F$ **consists of element-wise mappings**. *For all $b \in B, k \in dim(\mathbf{x}_b^{(m)})$,*

$$x_b^{(m)}(k) = F_{bk}(v_{a_1}^{(m)}(k), v_{a_2}^{(m)}(k), ...., v_{a_p}^{(m)}(k)) \tag{1}$$

($C_2$) **Non-redundant**. *The subgraph of the data-flow graph consisting of layers in $A$, $B$, and the connections between them form a single component.*

($C_3$) **Completeness**. *There do not exist sets $A', B' \subset [L]$ and a collection of functions $F'$ defined by the data flow graph of the network where $A \subseteq A'$ and $B \subseteq B'$ and either $A \neq A'$ or $B \neq B'$ such that $<A', B', F'>$ satisfies conditions ($C_1$) and($C_2$).*

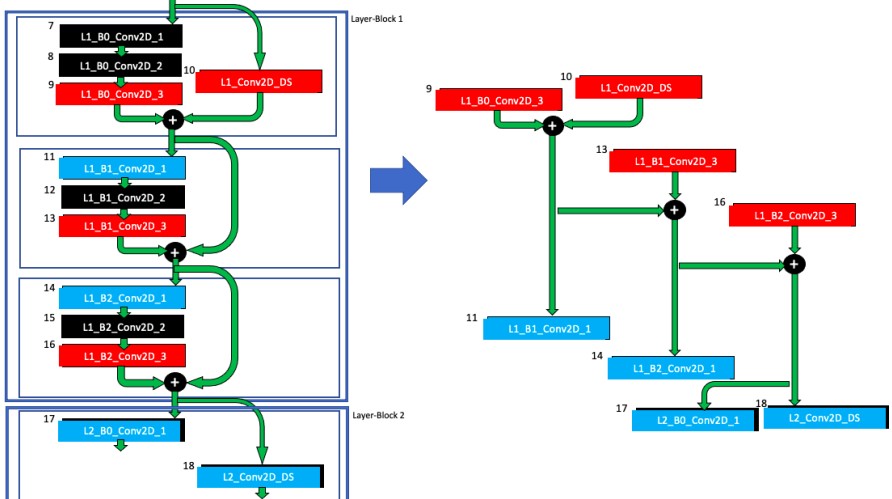

Figure 2: An instance of a DFC in ResNet-50.

We provide discussion for conditions $(C_2)$ and $(C_3)$ of Definition 1 in Appendix B.

**Some terminology for DFCs**. For a DFC $< A, B, F >$ denoted by $\tau$, we call the layers in $A$ the *feed-in* layers since their outputs are used as inputs by layers in $B$ post transformations governed by $F$. Consequently, we call the layers in $B$ the *feed-out* layers. Additionally, as a consequence of $(C_1)$, it is the case that for all $a \in A$ and $b \in B$, $dim(\mathbf{v}_a^{(m)}) = dim(\mathbf{x}_b^{(m)}) = n(\tau)$. We call $n(\tau)$ as the *cardinality of coupling*.

**DFCs in CNNs**. The notion of a DFC for a neural network with FC layers can be extended to a CNN by altering the element-wise property of transformations in $F$ to channel-wise.

**Example of a DFC in ResNet-50.** Consider the subnetwork of ResNet-50, as shown on the left in Figure 2. For simplicity, we only show the convolutional layers and residual additions from the data flow graph of the subnetwork. Each convolutional layer in the figure has its assigned unique integer on the top left corner besides its corresponding rectangular box in the diagram. This example focuses on the DFC involving channel coupling due to residual additions. The DFC $\tau$ is $< A, B, F >$, where $A = \{9, 10, 13, 16\}$, and $B = \{11, 14, 17, 18\}$. It is easy to infer the DFC's collection of functions $F$ from the diagram. However, for completeness, we show $F_{11,k}$.

$$F_{11,k}(\mathbf{v}_9^{(m)}(k), \mathbf{v}_{10}^{(m)}(k)) = ReLU(BN_{9,k}(\mathbf{v}_9^{(m)}(k))) + ReLU(BN_{10,k}(\mathbf{v}_{10}^{(m)}(k))) \quad (2)$$

where $BN_{l,k}(\mathbf{x}) = \frac{x - \mu_k^l}{\sigma_k^l} \odot \gamma_k^l + \beta_k^l$. Here, $BN_{l,k}$ denotes the batchnorm transformation applied on the $k^{th}$ channel output from the $l^{th}$ layer of the neural network.

**Properties of DFCs**. Note that in Figure 2, the tuple $< A, B, F >$ with $A = \{11\}$ and $B = \{12\}$ and $F$ capturing the associated transformation satisfies the definition of a DFC. Thus, the notion of consecutive layers, as in single-branch networks, is a special case of a DFC. Thus, DFCs simultaneously capture the transformational effect of coupled and non-CCs during the forward pass. Moreover, it is easy to see that a network can be divided into a collection of DFCs while preserving the overall transformation it produces.

## 4    GROUPED SALIENCIES

In this section, we investigate whether saliencies can be assigned to CCs in isolation using existing saliency mechanisms. To do so, we define Maximum Score Disagreement, which quantifies the disagreement or inconsistency in saliency rankings. When computed in isolation for each feed-in layer, our experiments suggest inconsistency among saliency ranks assigned to corresponding channels of a DFC. Thus, we propose *Grouped Saliencies* to rank channels of a DFC.

**Saliency mechanisms under consideration**. Broadly, structured pruning algorithms compute saliencies of non-coupled channels and discard the *lowest-ranked* channels first (Hoefler et al., 2021). Saliency scoring mechanisms exist for structured pruning that use statistics of the feed-in

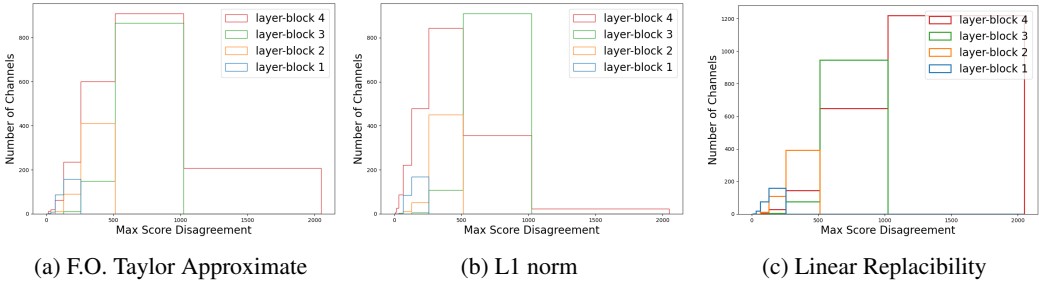

(a) F.O. Taylor Approximate       (b) L1 norm       (c) Linear Replicability

Figure 3: Plots illustrating disagreement of scores among the feed-in layers for gauging the importance of channels for a ResNet-50 model trained on CIFAR-100 dataset. Each plot is a histogram of the MSD values for the three saliency scoring mechanisms. A DFC with a particular downsampling layer in its feed-in layers is uniquely determined by the layer block containing it in the ResNet-50 architecture. The DFC in figure 2 is called layer-block-1 for this experiment. DFC named layer-block-1, layer-block-2, layer-block-3, and layer-block-4 have a cardinality of coupling equal to 256, 512, 1024, and 2048 respectively.

layer (Joo et al., 2021; Li et al., 2017; Molchanov et al., 2019). We use these saliencies to illustrate the necessity of grouped saliencies. Molchanov et al. (2019) propose to gauge the saliency of a filter in a convolutional layer by measuring the first-order Taylor approximation error attained on discarding the filter. Li et al. (2017) propose to use the L1 norm of the weights in the filter to gauge the corresponding channel's saliency. Joo et al. (2021) measure the saliency of a filter by measuring how linearly dependent the filter is on other filters in the layer. To capture the variation of ranks assigned by various feed-in layers of a DFC, we define the Maximum Score Disagreement (MSD) as follows.

**Definition 2** *Maximum Score Disagreement. For a DFC $< A, B, F >$, denoted by $\tau$, let $rank_a(k)$ denote the rank assigned by the feed-in layer $a$ to channel $k$ using a saliency scoring mechanism. We then define the Maximum Score Disagreement for this channel $k$ as*

$$MSD_\tau(k) = \max_{a,b \in A, a \neq b} |rank_a(k) - rank_b(k)| \tag{3}$$

**Discussion**. Given a DFC $\tau = < A, B, F >$, we compute a saliency score for each filter of all the feed-in layers separately, thus giving us a rank for each filter in each feed-in layer from the set $[n(\tau)]$. We say that the saliency mechanism is *consistent* or in *agreement* if $MSD_\tau(k) = 0 \; \forall k \in [n(\tau)]$. When under agreement, $rank_a(k) = rank_b(k) \; \forall k \in [n(\tau)]$, and $a, b \in A$. This means that all feed-in layers agree regarding the order in which the corresponding channels should be pruned. However, if $MSD_\tau(k)$ is large for some $k \in [n(\tau)]$, there is a disagreement among at least two layers regarding the importance of channel $k$.

**Experiments**. We perform three experiments to investigate the agreement among saliencies computed in isolation through the MSD values. We use the three saliency scoring mechanisms mentioned above in this section. We use all DFCs consisting of coupled channels arising from residual connections in ResNet-50 trained on the CIFAR-100 dataset (MIT License) for this experiment (training specifications in section F of the supplementary material).

**Observation**. Consider layer-block-4 in the histograms in figure 3c. The most frequent bin for this particular histogram lies in the MSD Range of 1000-2000. This shows significant disagreements among saliency ranks computed in isolation by the feed-in layers. Similarly, we can see that the disagreement for all three importance measures is significantly high for all histograms in all plots of figure 3. Similar trends also arise for ResNet-50 trained on CIFAR-10 and ImageNet datasets.

**Grouped Saliencies**. These observations show that we need to jointly consider coupled elements of a DFC to infer their saliencies. *We call saliencies that measure the importance of channels in a DFC using at least one of all the feed-in layers, all the feed-out layers, and the entire collection of functions $F$ as grouped saliencies.*

In the following section, we propose an algorithm that computes a grouped saliency using all three elements of the triple of a DFC as detailed in Definition 1.

## 5 A DATA FLOW DRIVEN DATA FREE GROUPED SALIENCY BASED ON THE RECONSTRUCTION ERROR OF INTERMEDIATE OUTPUTS

In this section, we propose an Algorithm called BGSC[1] to compute the saliency of all neurons in a DFC. We begin by describing the preliminaries for the Algorithm. We then describe the desired objective function to measure our saliency. Finally, through Theorem 1, we show that the saliencies computed using the BGSC Algorithm upper bound the desired objective function.

**Setup**. Consider a neural network with the DFC $< A, B, F >$ denoted by $\tau$ for which $\mathbf{u}_a$, $\mathbf{v}_a$ denote the input to and the output of layer $a \in A$, and by $\mathbf{x}_b$, $\mathbf{y}_b$ denote the same for layer $b \in B$. Let $P_{ba}$ denote the set of all paths from layer $b \in B$ to layer $a \in A$ in the backwards graph of the network. We aim to remove less important neurons from $\tau$. On removing a neuron from $\tau$, the output of the feed-out layers in $B$ may change. Thus, our goal is to select a neuron whose removal causes the least perturbation in the output across all feed-out layers of $\tau$.

**Measuring Saliencies**. Let $\mathbf{s} \in \{0, 1\}^{n(\tau)}$ be a mask, such that $\|\mathbf{s}\|_1 = n(\tau) - 1$. Here, setting $s(k) = 0$ for any $k \in [n(\tau)]$ is equivalent to pruning the $k^{th}$ neuron from $\tau$. Thus, to infer the least salient neuron in $\tau$, we would want to solve the following optimization problem.

$$\min_{k \in [n(\tau)]} \sum_{b \in B} \mathcal{OPT}(b) \ \ s.t. \ \ \|\mathbf{s}\|_1 = n(\tau) - 1, s(k) = 0 \quad (4)$$

where $\mathcal{OPT}(b) = \|\mathbf{W}_b \mathbf{x}_b - \mathbf{W}_b(\mathbf{x}_b \odot \mathbf{s})\|_1$ is the change in output of layer $b \in B$ on applying the mask $\mathbf{s}$.

---

**Algorithm 1** BGSC: Backwards Graph based Saliency Computation

---

**Input:** A DFC $\tau = < A, B, F >$, the backwards graph $G$
**Output:** List $Sal$.              $\triangleright$ $Sal(k)$ is saliency of $k^{th}$ neuron
 1: $Sal(k) \leftarrow 0$ for all $k \in n(\tau)$
 2: **for** each $a \in A$, $b \in B$ **do**
 3:      **for** each path $\pi$ between $b$ and $a$ in $G$ **do**
 4:          $\mathbf{acc} = |\mathbf{W}_b|^T \mathbf{e}$
 5:          **for** each node $\nu$ in $\pi$ **do**
 6:              **if** $\nu$ performs residual addition **then**
 7:                  Do nothing.
 8:              **else if** $\nu$ performs a Lipschitz continuous element-wise transformation **then**
 9:                  Find $\mathbf{C}$: matrix consisting tightest Lipschitz constants for the transformation.
10:                  $\mathbf{acc} = \mathbf{C}.\mathbf{acc}$
11:      **for** all $k \in [n(\tau)]$ **do**
12:          $\mathbf{s} \longleftarrow s(k) = 0, \ s(j) = 1 \forall \ j \in [n(\tau)] \setminus \{k\}$
13:          $\mathbf{acc}_{ba}^\pi(k) = |\mathbf{W}_a^T|(\mathbf{e}' - \mathbf{s}) \odot \mathbf{acc}$
14:          $Sal(k) = Sal(k) + \|\mathbf{acc}_{ba}^\pi(k)\|_1$

---

**Overview**. The BGSC Algorithm traverses through all paths in the backwards graph that exists between any pair of feed-out and feed-in layers of the DFC under consideration to compute the saliency of neurons. For each path, the Algorithm accumulates scores for each neuron. The saliency of a neuron is then obtained by summing up the scores accumulated from every path. This is shown in line 14 of Algorithm 1. While traversing each path $\pi$, the accumulated score is initialized as shown in line 4 of the Algorithm. Then as we traverse the backwards graph along path $\pi$ from the feed-out layer, we augment the accumulated score at every node depending on the operation it performs, as depicted in lines 7 and 10 of the Algorithm. Once we reach the feed-in layer, we perform one last augmentation to the accumulated score as depicted on line 13 of the Algorithm.

**Theorem 1** *Suppose $\tau = < A, B, F >$ is a DFC as defined in Definition 1. Let $\mathbf{acc}_{ba}^\pi$ be as computed in Algorithm 1 for all $a \in A, b \in B$, and $\pi \in P_{ba}$. Then,*

$$\mathcal{OPT}(b) \leq \sum_{a \in A} \sum_{\pi \in P_{ba}} (\mathbf{acc}_{ba}^\pi(k))^T |\mathbf{u}_a| \ \ \forall b \in B \quad (5)$$

*Proof of Theorem 1* is presented in Section C of the Appendix.

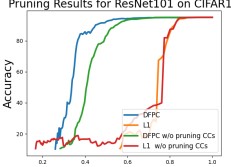 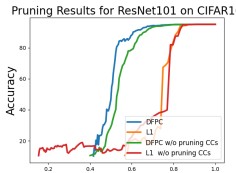 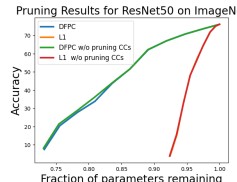 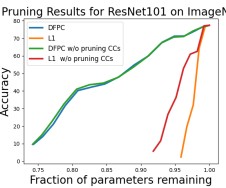

(a) Acc. vs. params for ResNet101 on CIFAR10    (b) Acc. vs. FLOPs for ResNet101 on CIFAR10    (c) Acc. vs. params for ResNet50 on ImageNet    (d) Acc. vs. params for ResNet101 on ImageNet

Figure 4: Figures comparing Accuracy (Acc.) versus Sparsity (Sp.) (through FLOPs or parameters) for DFPC and L1-norm-based pruning strategies under the data-free regime. Additionally, the plots also compare Acc. vs. Sp. when we choose to prune CCs vs. when we don't. In each figure, the blue and the green plots denote Acc. vs. Sp. for DFPC when we choose to prune CCs, and when we don't choose to prune CCs, respectively. Similarly, the orange and the red plots denote Acc. vs. Sp. for L1 when we choose to prune CCs, and when we don't choose to prune CCs, respectively.

From Theorem 1, we have

$$\sum_{b \in B} \mathcal{OPT}(b) \leq \sum_{a \in A, b \in B} \sum_{\pi \in P_{ba}} (\mathbf{acc}_{ba}^{\pi}(k))^T |\mathbf{u}_a| \leq \gamma \sum_{a \in A, b \in B} \sum_{\pi \in P_{ba}} \|\mathbf{acc}_{ba}^{\pi}(k)\|_1 \qquad (6)$$

Here, since we do not have access to the $\mathbf{u}_a$s, and we know that the pixel values of an input image are bounded, we define $\gamma = \max_{a \in A, I}\{\|u_a^{(I)}\|_\infty\}$. $u_a^{(I)}$ denotes the value of $u_a$ on feeding input $I$ to the network. Here, the maximization over $I$ denotes the maximization over the set of all possible images. Thus, we infer the saliency of a neuron $k$ in $\tau$ by

$$Sal_\tau(k) = \sum_{a \in A, b \in B} \sum_{\pi \in P_{ba}} \|\mathbf{acc}_{ba}^{\pi}(k)\|_1. \qquad (7)$$

**Time complexity of Algorithm 1**: Let $n$ be the number of nodes in the subgraph of the backwards graph consisting of the feed-out layers, the feed-in layers, and the connections between them for a DFC $\tau$. Also, let $P = \cup_{a \in A, b \in B} P_{ba}$ denote the set of all paths between the feed-in and feed-out layers of $\tau$ in the backwards graph of the network. If $\gamma_A = \max_{a \in A} dim(\mathbf{u}_a)$, and $\gamma_B = \max_{b \in B} dim(\mathbf{x}_b)$ then, the *time complexity of BGSC Algorithm is* $\mathcal{O}\{n(\tau).|P|.[\gamma_B + n(\tau).(n + \gamma_A)]\}$.

**BGSC Algorithms for CNNs**. The BGSC algorithm is defined here for multi-branched neural networks. However, while it is computationally expensive for CNNs, it is embarrassingly parallel. In Appendix D, we describe a parallelized implementation of the BGSC algorithm that is faster to execute. We use this parallelized version for our experiments.

## 6 PRUNING EXPERIMENTS

In this section, we present the results of our pruning experiments obtained using DFPC on CNNs. Since our work, to the best of our knowledge, is the first to adopt data-free pruning to prune coupled channels (CCs), we baseline our work against an extension of the L1-norm based saliency scores(Li et al., 2017) (similar to Gao et al. (2019)) and random pruning. In Appendix E, we show how we measure these saliencies for CCs. Moreover, to strengthen the experiments, we baseline against structured pruning algorithms in the data-driven regime on the ImageNet dataset. We also show that decreases in FLOPs does not yield a similar decrease in inference time, as also noted in Yang et al. (2018); Liu et al. (2021), highlighting the importance of pruning CCs. Details of experiments and the ablation studies are presented in Appendix F.

**Compute Platform.** Appendix A specifies the platform used for inference time measurements.

**Data-Free Experiments**. Experiments are performed on ResNet-50/101, MobileNet-v2, and VGG16 for the CIFAR-10/100 datasets. We also present results on ResNet-50/101 for the ImageNet dataset.

**Results of Data-Free Experiments**. Figures 4a, 4c, 4d, and more in Appendix F, show that DFPC consistently outperforms L1-based scores for a given sparsity budget whether we choose to prune CCs or not. For CIFAR-10/100 datasets, the performance of the L1-based saliency score is quite similar whether we chose to prune CCs or not, whereas pruning CCs with DFPC outperforms

DFPC when CCs are ignored, in terms of both FLOPs and parametric sparsity. For the ImageNet dataset, both DFPC and L1-based pruning perform similarly whether CCs are pruned or not. Finally, as noted in Tables 1 and 5, both accuracies and inference times improve as we prune CCs. Our experiments suggest that generally, DFPC outperforms the L1-based saliency score in terms of inference time gained for a particular drop in accuracy.

**Data-Driven Experiments and Results**. For comparison with contemporary work that finetune models after pruning, we present our results in Table 2. On a GPU, for a 0.2% accuracy drop, DFPC(30) attains an inference time speedup of 1.53x, similar to that of Greg-2(Wang et al., 2021), but with 2% higher test accuracy. Additionally, for an accuracy drop of 2.3%, similar to GReg-2, DFPC(54) attains a 2.28x speedup which is 49% higher than the nearest basline (GReg-2) on GPUs, and 47.1% faster than the nearest baseline (ThiNet-30) on our CPU platform. Moreover, DFPC attains a 52% higher speedup for the same reduction in FLOPs on GPUs, and 60% improvement on CPUs, over GReg-2 and OTO.

Table 1: Pruning Results without using the training dataset and no finetuning on CIFAR-10. RN is an abbreviation for ResNet; CP denotes if we choose to prune coupled channels; RF denotes the reduction in FLOPs; RP denotes the reduction in parameters; ITS denotes inference time speedup.

| Model Name | CP? | Acc-1(%) | RF | RP | ITS(CPU) | ITS(GPU) |
|---|---|---|---|---|---|---|
| Unpruned RN-50 | - | 94.99 | 1x | 1x | 1x | 1x |
| Random pruned RN-50 | No | 90.51 | 1.09x | 1.10x | 1.06x | 1.08x |
| L1-norm prunedLi et al. (2017) RN-50 | No | 88.33 | 1.38x | 1.32x | 1.16x | 1.17x |
| DFPC pruned RN-50 | No | 89.95 | **1.44**x | **1.82**x | **1.22**x | **1.22**x |
| Random pruned RN-50 | Yes | 88.39 | 1.09x | 1.09x | 1.13x | 1.11x |
| L1-norm prunedLi et al. (2017) RN-50 | Yes | 90.87 | 1.28x | 1.20x | **1.62**x | 1.31x |
| DFPC pruned RN-50 | Yes | 90.25 | **1.46**x | **2.07**x | 1.58x | **1.36**x |
| Unpruned RN-101 | - | 95.09 | 1x | 1x | 1x | 1x |
| Random pruned RN-101 | No | 90.35 | 1.15x | 1.14x | 1.14x | 1.06x |
| L1-norm prunedLi et al. (2017) RN-101 | No | 87.59 | 1.22x | 1.22x | 1.21x | 1.09x |
| DFPC pruned RN-101 | No | 89.80 | **1.53**x | **1.84**x | **1.56**x | **1.25**x |
| Random pruned RN-101 | Yes | 90.18 | 1.08x | 1.08x | 1.33x | 1.14x |
| L1-norm prunedLi et al. (2017) RN-101 | Yes | 90.31 | 1.22x | 1.21x | 1.31x | 1.11x |
| DFPC pruned RN-101 | Yes | 90.14 | **1.64**x | **2.22**x | **1.66**x | **1.35**x |
| Unpruned VGG-19 | - | 93.50 | 1x | 1x | 1x | 1x |
| Random pruned VGG-19 | - | 90.09 | 1.11x | 1.11x | 1.34x | 1.16x |
| L1-norm prunedLi et al. (2017) VGG-19 | - | 90.05 | 1.30x | 1.96x | 1.76x | 1.31x |
| DFPC pruned VGG-19 | - | 90.12 | **1.68**x | **3.16**x | **1.95**x | **1.43**x |

Table 2: ResNet-50 for ImageNet with finetuning. The number $x$ inside the brackets $(x)$ in the Model Name column denotes the pruned model obtained after $x$ pruning iterations.

| Model Name | FLOP Reduction | Parameter Reduction | Top-1 Accuracy(%) | Speedup (GPU) | Speedup (CPU) | FLOP Reduction by Speedup (GPU) | FLOP Reduction by Speedup (CPU) |
|---|---|---|---|---|---|---|---|
| Unpruned | 1.00x | 1.00x | 76.1 | 1.00x | 1.00x | 1.00x | 1.00x |
| GReg-2Wang et al. (2021) | 3.02x | 2.31x | 73.9 | 1.53x | 1.36x | 1.97 | 2.22 |
| OTOChen et al. (2021) | 2.86x | 2.81x | 74.7 | 1.45x | 1.25x | 1.97 | 2.29 |
| **DFPC(30)** | 1.98x | 1.84x | **75.9** | 1.53x | **1.42**x | **1.29** | **1.39** |
| ThiNet-30Luo et al. (2017) | 3.46x | 2.95x | 71.6 | 1.50x | 1.38x | 2.31 | 2.51 |
| **DFPC(54)** | 3.46x | 2.65x | **73.8** | **2.28**x | **2.03**x | **1.51** | **1.70** |

## 7 DISCUSSION AND CONCLUSION

This work proposes a data-free method to prune networks with coupled channels to obtain a superior accuracy vs inference time trade-off. To do this, we propose data flow couplings that abstract the coupling of channels in a network. We also show the necessity of defining grouped saliencies. Finally, we provide an algorithm to compute grouped saliencies on DFCs based on the reconstruction error of the output of the feed-out layers. We also provide a parallelized implementation of BGSC for use with CNNs. The algorithm attains superior speedups in both the data-free and data-driven regimes against our baselines. Notably, in the data-driven regime, DFPC pruned ResNet-50 obtains up to 47.1% faster models for a 2.3% accuracy drop on the ImageNet dataset. In the future, we aim to develop pruning strategies robust enough to prune arbitrary networks and advance the goal of achieving faster inference times.

## ACKNOWLEDGEMENTS

The authors gratefully acknowledge AMD for their support. The authors also thank Ramaswamy Govindarajan (Professor, IISc), Himanshu Jain (IISc), Mehul Shankhapal (IISc), Raghavendra Prakash (AMD), and Ramasamy Chandra Kumar (AMD) for their insightful discussions pertaining to this paper.

The authors thank the reviewers for their valuable feedback which has helped us improve our work.

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

# APPENDIX

The appendix is structured as follows.

(a) In Appendix A, we specify the setup and the procedure used to measure the inference time of models for the pruning experiments performed throughout the manuscript.

(b) In Appendix B, we provide discussion on conditions $(C_2)$ and $(C_3)$ in the Definition of a Data Flow Coupling as defined in Section 3 of the manuscript.

(c) In Appendix C, we present the Proof of Theorem 1 as promised in Section 5 of the manuscript.

(d) In Section 6 of the main paper, we perform our experiments on CNNs. But, our definitions and derivations in Sections 3, and 5 consider neural networks with linear/fully-connected layers. In Section D, we discuss how to apply the BGSC Algorithm(Algorithm 1 of the manuscript) to CNNs to compute the saliencies of channels.

(e) As a part of our pruning experiments from Section 6 of the manuscript, we compare the efficacy of DFPC against two grouped saliencies extended from the L1-based and random saliency mechanisms in the data-free regime. Section E shows how we extended the said saliency mechanisms to grouped saliencies.

(f) In Section F, we state the experimental procedures and their results in detail for our pruning experiments presented in Section 6 of the manuscript.

## A  SPECIFICATIONS FOR INFERENCE TIME MEASUREMENTS

**Inference time measurements**. We define the time taken to inference a model on the test set as its inference time. Inference time for a given model is measured as follows in our experiments. The five epochs are warmups, and we discard their results. The inference time is now computed as the average of the next ten epochs. Shen et al. (2021) use a similar method to measure inference times. Inference time does not include the time taken to load data into memory.

## A.1 CPU Hardware

Table 3: Specifications of CPU hardware used for inference time measurements

| CPU Model Name | AMD EPYC 7763 64-Core |
| --- | --- |
| CPU(s) | 256 |
| Thread(s) per core | 2 |
| Core(s) per socket | 64 |
| Socket(s) | 2 |
| NUMA node(s) | 8 |
| CPU MHz | 2445.419 |
| L1d & L1i cache | 4 MiB |
| L2 cache | 64 MiB |
| L3 cache | 512 MiB |
| RAM | 1TB (DDR4, 3200 MT/s) |

The CPU inference time measurements performed as a part of the pruning experiments in Section 6 are performed using the OS Ubuntu 20.04.3 LTS with kernel 5.13.0-39-generic on the hardware specified in Table 3. The software stack used for inferencing consisted of Python 3.9.7, PyTorch 1.10.1, and Torchvision 0.11.2.

## A.2 GPU Hardware

The GPU inference time measurements performed as a part of the pruning experiments in Section 6 are performed using the OS Ubuntu 16.04.7 LTS with kernel 4.15.0-142-generic on the hardware specified in Table 3. The GPU is an NVIDIA 1080 Ti with CUDA 10.2 and a memory of 12GB. The software stack used for inferencing consisted of Python 3.9.7, PyTorch 1.10.1, and Torchvision 0.11.2.

## B Discussion for conditions ($C_2$) and ($C_3$) in Definition of a Data Flow Coupling

In this section, we discuss the requirement of conditions ($C_2$) and ($C_3$) in defining a Data Flow Coupling through examples. We begin by restating the definition and then providing examples that illustrate the importance of the two conditions.

**Setup**. Consider a neural network with $L$ FC layers where each FC layer is assigned a unique integer from the set $[L]$. Now, consider two sets of layers $A = \{a_1, a_2, ..., a_p\}, B = \{b_1, b_2, ..., b_q\}$ where $A, B \subset [L]$. Let $\mathbf{z}^{(m)}$ be an arbitrary input sample from the data set $\{\mathbf{z}^j\}_{j=1}^M$ that is fed to the network. Then, by $\mathbf{u}_a^{(m)}$, $\mathbf{v}_a^{(m)}$ denote the input to and the corresponding output of layer $a \in A$, and by $\mathbf{x}_b^{(m)}$, $\mathbf{y}_b^{(m)}$ denote the same for layer $b \in B$. Let $A, B$ be such that there exists a collection of functions $F$ defined by the data flow graph of the network. The input to any layer $b \in B$ is obtained through a map $\mathbf{F}_b : \mathbb{R}^{\sum_{a \in A} dim(\mathbf{v}_a^{(m)})} \to \mathbb{R}^{dim(\mathbf{x}_b^{(m)})} \in F$. $\mathbf{F}_b$ is a function of the outputs of layers $a \in A$. Let the function that gives the value of activation to the $k^{th}$ neuron in $\mathbf{F}_b$ be denoted by $F_{bk}$.

**DFC Definition**. The tuple $\tau = <A, B, F>$ is a data flow coupling if

($C_1$) $F$ **consists of element-wise mappings**. For all $b \in B, k \in dim(\mathbf{x}_b^{(m)})$,

$$x_b^{(m)}(k) = F_{bk}(v_{a_1}^{(m)}(k), v_{a_2}^{(m)}(k), ...., v_{a_p}^{(m)}(k)) \tag{8}$$

($C_2$) **Non-redundant**. The subgraph of the data-flow graph consisting of layers in $A$, $B$, and the connections between them form a single component.

($C_3$) **Completeness**. There do not exist sets $A', B' \subset [L]$ and a collection of functions $F'$ defined by the data flow graph of the network where $A \subseteq A'$ and $B \subseteq B'$ and either $A \neq A'$ or $B \neq B'$ such that $<A', B', F'>$ satisfies conditions ($C_1$) and ($C_2$).

**Discussion for Condition** ($C_2$). We include this condition to avoid including redundant channels in a DFC. Consider two DFCs in a network with the same cardinality of coupling and no layers in

common between the two DFCs. One might mistakenly club the two DFCs into one by taking the union of their feed-in and feed-out layers, respectively. Thus, if condition ($C_2$) were not present, the combination of the two DFCs would also become a DFC. This would create an undesired constraint to prune channels from both DFCs simultaneously.

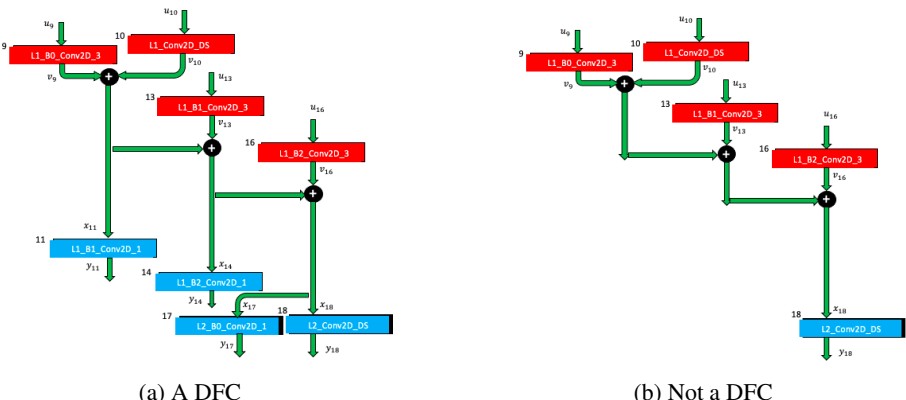

(a) A DFC                                        (b) Not a DFC

Figure 5: Figures illustrating the importance of the completeness condition in the definition of a Data Flow Coupling.

**Discussion for Condition** ($C_3$). This completeness condition ensures that none of the feed-in or the feed-out layers is left out when considering a DFC. Let us assume that the set of layers and transformations in Figure 5a satisfies the definition of a DFC. If condition ($C_3$) were not present, one could mistakenly not consider all feed-in or feed-out layers while considering this DFC. An example for such an error is shown in Figure 5b.

## C    PROOF OF THEOREM 1

In this section, we present the proof to Theorem 1 posited for the BGSC Algorithm in the main manuscript. We begin by setting up the mathematical preliminaries and re-stating the Theorem 1. Finally, we present our proof.

**Setup**. Consider a neural network with the DFC $< A, B, F >$ denoted by $\tau$ for which $\mathbf{u}_a$, $\mathbf{v}_a$ denote the input to and the corresponding output of layer $a \in A$, and by $\mathbf{x}_b$, $\mathbf{y}_b$ denote the same for layer $b \in B$. In $\tau$, each function $\mathbf{F}_b$ captures element-wise transformations from operations like batch-normalization, non-linearities, etc. Thus, we model $\mathbf{F}_b$ as a composite function. That is, $\mathbf{F}_b = \mathbf{f}_b^1(\mathbf{f}_b^2(...))$ where each $\mathbf{f}_b^t$ is an element-wise function of $\mathbf{v}_a$s. Let $P_{ba}$ denote the set of all paths from layer $b \in B$ to layer $a \in A$ in the backwards graph of the network.

**Assumption 1** *We assume that all functions $\mathbf{f}_b^t$ in $\tau$ map the additive identity of their domain to the additive identity of their co-domain and are Lipschitz continuous.*

**Optimization Problem**. Let $\mathbf{s} \in \{0, 1\}^{n(\tau)}$ be a mask, such that $\|\mathbf{s}\|_1 = n(\tau) - 1$. Here, setting $s(k) = 0$ for any $k \in [n(\tau)]$ is equivalent to pruning the $k^{th}$ neuron from $\tau$. Thus, to infer the least salient neuron in $\tau$, we want to solve the following optimization problem.

$$\min_{k \in [n(\tau)]} \sum_{b \in B} \mathcal{OPT}(b) \;\; s.t. \;\; \|\mathbf{s}\|_1 = n(\tau) - 1, s(k) = 0 \tag{9}$$

where $\mathcal{OPT}(b) = \|\mathbf{W}_b \mathbf{x}_b - \mathbf{W}_b(\mathbf{x}_b \odot \mathbf{s})\|_1$ is the change in output of layer $b \in B$ on applying the mask $\mathbf{s}$.

**Theorem 1**  *Let $\mathbf{acc}_{ba}^{\pi}(k)$ be as computed in Algorithm 1 for all $a \in A, b \in B$, and $\pi \in P_{ba}$. Then,*

$$\mathcal{OPT}(b) \leq \sum_{a \in A} \sum_{\pi \in P_{ba}} (\mathbf{acc}_{ba}^{\pi}(k))^T |\mathbf{u}_a| \;\; \forall b \in B \tag{10}$$

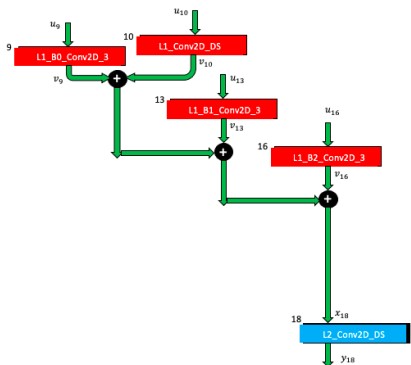

Figure 6: Focussing on layer 18, a feed-out layer in a DFC.

**Proof of Theorem 1** We focus on one feed-out layer, $b$ of the DFC $\tau$. For instance, consider layer 18 in Figure 6. Consider $\mathcal{OPT}(b)$. Let $\mathbf{e}, \mathbf{e}'$ be vectors whose element are all 1s. The dimensions of $\mathbf{e}, \mathbf{e}'$ should be clear from the context. We have

$$\mathcal{OPT}(b) = \mathbf{e}^T |\mathbf{W}_b \mathbf{x}_b - \mathbf{W}_b(\mathbf{x}_b \odot \mathbf{s})| \leq (|\mathbf{W}_b|^T \mathbf{e})^T . |(\mathbf{e}' - \mathbf{s}) \odot \mathbf{x}_b| \quad (11)$$

We now start unfolding the cascadation of functions that obtain $\mathbf{x}_b$ from the $\mathbf{u}_a$s to prove the theorem. Let us define the accumulated score, $\mathbf{acc}$, as the vector on the left in the inner-product of the right-most term of 11. That is,

$$\mathbf{acc} = |\mathbf{W}_b|^T \mathbf{e} \quad (12)$$

Let us now perform a case-wise analysis on $\mathbf{f}_b^t$. Let the accumulated score until unfolding level $t$ be $\mathbf{acc}$.

(a) *Residual Connection*: If $\mathbf{f}_b^t = \mathbf{f}_{b1}^{t+1} + \mathbf{f}_{b2}^{t+1}$ where both $\mathbf{f}_{b1}^{t+1}$ and $\mathbf{f}_{b2}^{t+1}$ are element-wise functions on $\mathbf{v}_b$s. Then, we have

$$\mathbf{acc}^T . |(\mathbf{e}' - \mathbf{s}) \odot \mathbf{f}_b^t| \leq \mathbf{acc}^T . |(\mathbf{e}' - \mathbf{s}) \odot \mathbf{f}_{b1}^{t+1}| + \mathbf{acc}^T . |(\mathbf{e}' - \mathbf{s}) \odot \mathbf{f}_{b2}^{t+1}| \quad (13)$$

(b) *Elementwise Lipschitz continuous transformation*: When $\mathbf{f}_b^t$ is Lipschitz continuous, there exists a constant $C_k$, for each $f_b^t(k)$, such that $|f_b^t(k)(r) - f_b^t(k)(s)| \leq C_k |r - s|$ for any two scalars $r$ and $s$ in the domain of $f_b^t(k)$. Then, from Assumption 1, we have

$$\mathbf{acc}^T . |(\mathbf{e}' - \mathbf{s}) \odot \mathbf{f}_b^t(\mathbf{f}_b^{t+1})| = \mathbf{acc}^T . |\mathbf{f}_b^t(\mathbf{f}_b^{t+1}) - \mathbf{f}_b^t(\mathbf{s} \odot \mathbf{f}_b^{t+1})|$$
$$\leq \mathbf{acc}^T . |\mathbf{C}|\mathbf{f}_b^{t+1} - \mathbf{s} \odot \mathbf{f}_b^{t+1}|| = (\mathbf{C}.\mathbf{acc})^T |(\mathbf{e}' - \mathbf{s}) \odot \mathbf{f}_b^{t+1}| \quad (14)$$

where $C$ is a diagonal matrix with $C_k$ as its $k^{th}$ diagonal element. Thus, the new accumulated score is

$$\mathbf{acc}_{new} = \mathbf{C}.\mathbf{acc}. \quad (15)$$

Additionally, to generate a tighter upper bound for equations 11, we use the smallest constant $C_k$ that satisfy Lipschitz continuity for $f_b^t(k)$.

As one unfolds $\mathbf{F}_j$ to attain upper bounds on $\mathcal{OPT}(b)$ using 13 and 15 in a DFC $< A, B, F >$, we are guaranteed to attain a situation where either of $\mathbf{f}_{b1}^{t+1}, \mathbf{f}_{b2}^{t+1}, \mathbf{f}_b^{t+1}$ performs no transformation on its only input $\mathbf{v}_a$ for some $a \in A$. Denote by $\mathbf{a\hat{c}c}_{ba}^\pi$ the score accumulated until now by unfolding transformations from $b$ to $a$ along the path $\pi$ in the backwards graph of the network. This condition should occur by the construction of the network. From here, we perform one more step of unfolding, where we have

$$\mathbf{a\hat{c}c}_{ba}^\pi . |(\mathbf{e}' - \mathbf{s}) \odot \mathbf{v}_a| \leq \{|\mathbf{W}_a^T|(\mathbf{e}' - \mathbf{s}) \odot \mathbf{a\hat{c}c}_{ba}^\pi\}^T . |\mathbf{u}_a| \quad (16)$$

Finally, the accumulated score for one path $\pi$ in the backwards graph from $b$ to $a$ is $\mathbf{acc}_{ba}^\pi(k) = |\mathbf{W}_a^T|(\mathbf{e}' - \mathbf{s}) \odot \mathbf{a\hat{c}c}_{ba}^\pi.\square$

**Gist of proof**: The value $\mathbf{acc}_{ba}^\pi(k).|\mathbf{u}_a| = |\mathbf{W}_a^T|(\mathbf{e}' - \mathbf{s}) \odot \mathbf{a\hat{c}c}_{ba}^\pi.|\mathbf{u}_a|$ for a feed-in layer $a$ and a feed out layer $b$ of a DFC measures an upper bound to the change in value of the output of $b$ if we were to make output value of layer $a$ to be zero for channel $k$ and only for the path $\pi$. Thus, to measure the upper bound of joint perturbations of outputs of all feed-out layers on removing a channel $k$, we take a summation of these upper bounds across all paths $\pi$ from between the feed-out and the feed-in layers.

## D    APPLYING BGSC TO CNNS

Across definitions and derivations in Sections 3, 4, and 5, we consider networks with fully-connected layers as the only layers that do not perform element-wise transformations. But, we demonstrate the efficacy of our method through experiments on CNNs in Section 6. CNNs consist of convolutional layers which do not perform element-wise transformations. In this Section, we show an equivalent linear layer for any convolutional layer and how to use the BGSC Algorithm to compute the saliencies of channels in DFCs that consist of convolutional layers.

### D.1    LINEAR LAYER EQUIVALENT TO A CONVOLUTIONAL LAYER

A convolutional layer with $m$ input and $n$ output channels consists of $n$ filters and $m$ kernels per filter. Let the $i^{th}$ filter be denoted by the weight tensor $\mathcal{W}_i \in \mathbb{R}^{m \times K \times K}$ for all $i \in [n]$ where $K \times K$ is the size of the kernel. Let the $j^{th}$ kernel in the $i^{th}$ filter be denoted by the matrix $\mathcal{W}_{ij} \in \mathbb{R}^{K \times K}$ for all $j \in [m]$. Assuming the bias terms to be zero, if the $j^{th}$ input channel and the $i^{th}$ output channel are denoted as $\mathcal{I}_j$ and $\mathcal{O}_i$ respectively then for all $i \in [n]$,

$$\mathcal{O}_i = \sum_{j \in [m]} \mathcal{W}_{ij} \circledast \mathcal{I}_j \tag{17}$$

where $\circledast$ denotes the convolutional operation. Let us denote by $\mathcal{O}_{ij} = \mathcal{W}_{ij} \circledast \mathcal{I}_j$. Then the $(p, q)^{th}$ element of the matrix $\mathcal{O}_{ij}$ is given by

$$\mathcal{O}_{ij}(p, q) = \sum_{r=0}^{K-1} \sum_{s=0}^{K-1} \mathcal{W}_{ij}(p, q).\mathcal{I}_j(p + r, q + s). \tag{18}$$

This is a linear transformation. Thus, we can find an equivalent matrix for a convolutional operation. Thus, if $\hat{\mathcal{I}}_j$ and $\hat{\mathcal{O}}_{ij}$ denote the flattened vectors corresponding to the matrices $\mathcal{I}_j$ and $\mathcal{O}_{ij}$ respectively, then there exists a matrix $\hat{\mathcal{W}}_{ij}$ such that $\hat{\mathcal{O}}_{ij} = \hat{\mathcal{W}}_{ij}\hat{\mathcal{I}}_j$. If $\hat{\mathcal{O}}_i$ denotes the flattened vector corresponding to the matrix $\mathcal{O}_i$, we have $\hat{\mathcal{O}}_i = \sum_{j \in [m]} \hat{\mathcal{W}}_{ij}\hat{\mathcal{I}}_j$. Then, we can write the transformation of a convolutional layer through a linear layer as follows.

$$\begin{pmatrix} \hat{\mathcal{O}}_1 \\ \hat{\mathcal{O}}_2 \\ ... \\ \hat{\mathcal{O}}_n \end{pmatrix} = \begin{pmatrix} \hat{\mathcal{W}}_{11}, \hat{\mathcal{W}}_{12}, ..., \hat{\mathcal{W}}_{1m} \\ \hat{\mathcal{W}}_{21}, \hat{\mathcal{W}}_{22}, ..., \hat{\mathcal{W}}_{2m} \\ ... \\ \hat{\mathcal{W}}_{n1}, \hat{\mathcal{W}}_{n2}, ..., \hat{\mathcal{W}}_{nm} \end{pmatrix} \begin{pmatrix} \hat{\mathcal{I}}_1 \\ \hat{\mathcal{I}}_2 \\ ... \\ \hat{\mathcal{I}}_m \end{pmatrix} \tag{19}$$

**Finding weight matrix for the equivalent linear layer**. A convolutional layer has multiple configurations, such as padding, strides, and dilation. One way to computationally find the equivalent linear layer to a convolutional layer in the presence of all such configurations is to emulate the convolution operation. During the emulation, fill the equivalent linear layer's weight matrix if an input contributes to the computation of the output by the corresponding weight in the corresponding kernel.

**Observation**. The equivalent linear layer's weight matrix is sparse (consisting of many 0s). Additionally, the weight matrix stores $m.n.I_x.I_y.O_x.O_y$ elements, where $I_x, I_y$ and $O_x, O_y$ represent the dimensions of $\mathcal{I}_j$ and $\mathcal{O}_i$ respectively. This number can grow very large very quickly.

**Using the BGSC Algorithm for CNNs**. To measure the saliencies of channels in a DFC $\tau$ of a CNN, we first need to think in terms of channels. Instead of element-wise transformations, the focus shifts to channel-wise transformations. An output channel of a channel-wise transformation depends only on the corresponding input channel. The shape of the output and input channels need not be the same; however, the number of input and output channels must be the same in a channel-wise transformation. Additionally, the mask **s** is changed. Consider a convolutional layer in the set of feed-in layers. If we want to prune the $i^{th}$ channel, the mask is such that $s(j) = 0$ for all $(i - 1)O_xO_y < j \le iO_xO_y$ and $s(j) = 1$ for all other $j$s.

### D.2    PARSING THROUGH CHANNEL-WISE OPERATIONS OF A CNN IN BGSC ALGORITHM

In this Section, we discuss how to parse through various channel-wise transformations in the BGSC Algorithm to compute the saliencies of channels in a DFC consisting of convolutional layers. Note that all element-wise transformations are also channel-wise transformations. But, the converse does not hold.

### D.2.1 ReLU Operation

The tightest Lipschitz constant for a ReLU function is 1. This clearly is the case since $|\max\{0,x\} - \max\{0,y\}| \leq |x - y|$ for any $x, y \in \mathbb{R}$. Thus, the matrix $\mathbf{C}$ consisting of the tightest Lipschitz constants [line 9 of BGSC Algorithm(1)] for a ReLU operation is an identity matrix.

### D.2.2 Batch normalization (2D)

For channel $k$, a batch norm layer linearly transforms each element of the $k^{th}$ channel of the input, $x$, as $\frac{x-\mu_k}{\sigma_k}.\gamma_k + \beta_k$ where $\mu_k, \sigma_k, \gamma_k, \beta_k$ are the parameters in a batch norm layer. Thus, the $(i,i)^{th}$ element of the matrix $\mathbf{C}$ is $\left|\frac{\gamma_k}{\sigma_k}\right|$ where $k$ denotes the channel the $i^{th}$ input/output element belongs to.

### D.2.3 Max-pooling, Average-pooling (2D)

For each feature-map corresponding to every input channel, the pooling operation operates on each patch of the feature-map to reduce their size. Max-pool computes the maximum value for each patch of a feature map to create the downsampled feature map. Average-pool computes the average value for each patch of a feature map to create the downsampled feature map.

Consider a pooling kernel of size $K_1 \times K_2$. We assume that for max-pooling, over a sufficiently large number of samples, each element is equally likely to be the maximum element in any patch of the image of size $K_1 \times K_2$. Thus, in the long run, the transformation by the max-pool and average-pool is equivalent to a convolutional layer whose specifications follow. If the number of channels input to the pooling layers is $m$, then the convolutional layer has $m$ input and output channels with filters such that for every filter $i \in [m]$, $\mathcal{W}_{ij}$ is a matrix with all its entries as $\frac{1}{K_1 K_2}$ if $j = i$ and 0 otherwise. The bias term is 0 for each channel, and the remaining configurations, like stride, padding, and dilation, remain the same as that of the pooling layer.

Now, from Section D.1, there exists an equivalent linear layer $l$ with weight matrix $W_l$ for the convolutional layer that is equivalent to the pooling layers. If the accumulated score is $\mathbf{acc}_{ba}^\pi$ until the BGSC Algorithm reaches node $l$. Then we update the score as

$$\mathbf{acc}_{ba}^\pi = W_l^T \mathbf{acc}_{ba}^\pi. \tag{20}$$

This is justified through the following inequality in the analysis presented in Section 5.

$$(\mathbf{acc}_{ba}^\pi)^T |\mathbf{W}_l \mathbf{f}_b^t - \mathbf{W}_l(\mathbf{f}_l^t \odot \mathbf{s})| \leq (|\mathbf{W}_l|^T \mathbf{acc}_{ba}^\pi)^T . |(\mathbf{e}' - \mathbf{s}) \odot \mathbf{f}_l^t| \tag{21}$$

### D.2.4 Adaptive Average Pooling (2D)

An adaptive average pooling performs average pooling. Here the pooling operation is specified by the shape of the output feature-map desired. Thus the kernel size for the layer is appropriately selected. Once the kernel size is identified, the methodology is the same as that of average-pooling(D.2.3).

### D.3 Miscellaneous Implementation Details

In this Section, we describe choices made while implementing BGSC to produce the results in Section 6. Moreover, we report the execution times for BGSC algorithms on our hardware.

### D.3.1 Reducing memory usage

Consider the second convolutional layer in VGG-19. It takes 64 channels of 32x32 images as input and produces an output of the same dimensions. From Section D.1, we know that the equivalent linear layer for this convolutional layer will require space to store $2^{32}$ floating point numbers. Assuming each number takes one byte of memory, the memory requirement for the weight matrix is already 4GB. This number jointly grows bi-quadratically with the dimensions of the input and output feature maps. Thus, to reduce this memory requirement, we use the sparse representation of matrices to represent the weight matrices corresponding to the equivalent linear layer.

### D.3.2 Reducing time to compute saliencies of channels in all DFCs in a network through parallelization

The time complexity of the BGSC Algorithm for a DFC is $\mathcal{O}\{n(\tau).|P|.[\gamma_B + n(\tau).(n + \gamma_A)]\}$. In a DFC, $\gamma_A, \gamma_B, n(\tau)$ are generally of the same order. So, we define $\gamma_{max} = \max\{\gamma_A, \gamma_B, n(\tau)\}$.

Then, we can write the time-complexity of BGSC Algorithm to be $\mathcal{O}\{\gamma_{max}^2.|P|.(n + \gamma_{max})\}$. We know that the $\gamma_{max}$ for a DFC with convolutional layers grows quadratically with respect to the dimensions of feature maps and linearly with the number of channels. Thus BGSC is quite computationally expensive. However, we reduce the time taken to execute BGSC Algorithm by parallelly computing the $\mathbf{acc}_{ba}^\pi$ for each path $\pi \in P$. Moreover, since saliency computation of two DFCs can be performed independently, we parallelly compute saliencies for channels of multiple DFCs of the network.

### D.3.3 PRACTICAL RUNTIME OF BGSC FOR RESNET-50 AND MOBILENET-V2 ARCHITECTURES.

For exposition, we presented the BGSC Algorithm in Section 5. This algorithm is computationally expensive if run sequentially as per the pseudocode in Algorithm 1. In an attempt to speedup saliency computation of a DFC using the BGSC Algorithm on CNNs for our experiments, we exploited the embarrassingly parallel nature of this algorithm. In an attempt to prototype this algorithm for ResNet-50/100, MobileNet-v2, and VGG-19, we were able to reduce the saliency computation time. However, there still remains scope for improvement which we discuss in this section in case one aims to deploy this algorithm in production. We also report the time it takes to compute saliencies for the ResNet-50 and MobileNet-v2 architectures.

**Execution times for BGSC Algorithm**. We now report the time taken to execute BGSC Algorithm on our CPU hardware (specified in Appendix A.1) for computing saliencies of channels in all DFCs in ResNet-50 and MobileNet-v2 when each method can call upto 10 threads (–num-processes argument in our code).

Table 4: Latency of executing BGSC algorithm to compute saliencies of all channels in specified networks when each method can call upto 10 threads.

| Model Name | Dataset | Execution Time |
|---|---|---|
| ResNet-50 | ImageNet | 38 minutes |
| ResNet-50 | CIFAR-10/100 | 12 minutes |
| MobileNet-v2 | CIFAR-10/100 | 68 seconds |

**Further improvements possible**. We now list further improvements to improve the latency of executing BGSC for all DFCs of a network. It is important to note that for the execution times measured for BGSC algorithm above, at one instant, the maximum number of DFCs that can be processed is also 10.

- In Python, multithreading is not truly possible due to GIL (Global Interpreter Lock), thus we use multiprocessing. There are associated overheads with multiprocessing that affect the execution time of the BGSC algorithm. Thus, dedicated effort to write code in C++ or CUDA may be benefical in reducing this overhead cost.

- Due to lack of support, we were unable to leverage parallelism for sparse matrix multiplication. Dedicated effort to parallelise sparse matrix multiplication can further reduce the time taken to execute the BGSC Algorithm.

## E EXTENDING L1-NORM BASED AND RANDOM SCORES TO PRUNE COUPLED CHANNELS

In this Section we demonstrate the usage of the two saliency scoring mechanisms, L1-norm and random, to prune coupled channels. These have been used as a benchmark to compare DFPC against in our Pruning Experiments(6).

Consider a CNN with $L$ convolutional layers. Let us assign each convolutional layer in the CNN a unique integer in $[L]$. Additionally, consider a DFC $< A, B, F >$ denoted by $\tau$ in the CNN.

### E.1 EXTENDING L1-NORM BASED SALIENCY SCORE

For a convolutional layer $l \in [L]$, Li et al. (2017) assign the $k^{th}$ channel a score of $\|\mathcal{W}_k^l\|_1$ where $\mathcal{W}_k^l$ denotes the weights of the $k^{th}$ filter in layer $l$. We extend this saliency score to a grouped saliency score as follows.

We assign a saliency score to channel $k \in [n(\tau)]$ as the sum of L1-norms of the corresponding filters across all feed-in layers. That is,

$$Sal_\tau(k) = \sum_{a \in A} \|\mathcal{W}_k^a\|_1 \ \ \forall k \in n(\tau). \tag{22}$$

This extension is similar to that proposed by Gao et al. (2019).

### E.2 EXTENDING RANDOM SALIENCY SCORE

Extending this saliency score is trivial. We assign each channel $k \in n(\tau)$ a number sampled from the uniform distribution between 0 and 1 as a saliency score. That is,

$$Sal_\tau(k) \sim \mathcal{U}[0, 1] \ \ \forall k \in n(\tau). \tag{23}$$

Here, $\mathcal{U}[a, b]$ denotes uniform distribution between scalars $a, b \in \mathbb{R}, a \leq b$.

## F EXPERIMENTS IN DETAIL AND ABLATION STUDIES

In this Section, we present a comprehensive version of our experiments that we presented in Section 6 of the main manuscript. We begin by presenting the experiments performed on the CIFAR-10 and CIFAR-100 datasets. Then, we present the experiments performed on the ImageNet dataset.

### F.1 CIFAR-10 AND CIFAR-100 EXPERIMENTS

**Experimental Setup.** We showcase our results using the CIFAR-10 and CIFAR-100 datasets (MIT License) and ResNet-50, ResNet-101, MobileNet-v2, and VGG-19. In these experiments, we maintain a data-free regime. Additionally, we use two settings for our experiments to show the effect of pruning coupled channels and fairly compare DFPC, L1, and random scores for ablation. In the first setting, we prune both the coupled and non-coupled channels in the network. In the second setting, we only prune the non-coupled channels. This helps us understand the gain obtained from pruning coupled channels. These experiments are carried out for three grouped saliencies: DFPC, L1, and Random. Moreover, these experiments are performed two times on ResNets. In this first set of experiments, we prune both coupled and non-coupled channels. But in the second set of experiments, we only prune the non-coupled channels.

**Pruning Procedure**. Once we obtain grouped saliencies $Sal_\tau(k)$ for each channel of every DFC in a network, we compare these scores globally to select the channel to prune among all DFCs. To prevent layer collapse, we add a check not to prune a channel if a DFC has a coupling cardinality of 1.

**Pretrained Models**. We train the models using SGD Optimizer with a momentum factor of $0.9$ and weight decay of $5 \times 10^{-4}$ for 200 epochs using Cosine Annealing step sizes with an initial learning rate of $0.1$.

**Tables 1, 5, and 6**. We produce these tables as follows. We prune 1% of the remaining channels at a time in the network and measure the top-1 accuracy of the pruned model. In these tables, we report the description of pruned models with accuracy closest to 90% for CIFAR-10 and 70% for CIFAR-100. For random saliencies, the tables report the average values obtained after three trials.

**Figures**. In figures 7- 13, we plot the results of our pruning experiments to show how accuracy varies with sparsities (with respect to FLOPs and parameters) when we choose to prune coupled channels for various strategies to gauge the importance of coupled channels.

### F.1.1 DISCUSSION OF EXPERIMENTAL RESULTS

From figures 7- 13, it is evident that DFPC outperforms L1 and Random grouped saliencies in accuracy versus sparsity charts for both sparsity in terms of parameters and FLOPs. The margin of outperformance is significantly higher when pruning coupled channels. We observe that this superiority arises due to the occurrence of more pruning iterations to obtain a similar accuracy drop. Additionally, the gap of outperformance is reduced when sparsity is considered with respect to FLOPs. For all cases, but two, DFPC results in a pruned model with faster inference time despite similar accuracies. It is for ResNet-50 and MobileNet-v2 trained on CIFAR-10 that L1-norm-based grouped saliency produces a pruned model with faster inference time when pruning coupled channels on our CPU platform. Additionally, L1-norm-based grouped saliency performs

similarly in terms of accuracy versus sparsity charts whether we chose to prune coupled channels or not. However, in the same regime, DFPC performs slightly better when pruning coupled channels. Thus, by looking at the trends in figures 7-12 and tables 1 and 5, it is the case that in general, for a given accuracy, both sparsity (in terms of FLOPs and number of parameters) and inference time speed-ups when pruning coupled channels are at least as good as when not pruning them.

To conclude, we were able to prune models without having access to the training data set or any statistics derived from it. We did not use fine-tuning either. We showed that our proposed method almost always improves performance in terms of sparsity and inference time speedups as opposed to readily-available approaches to gauge saliencies of coupled channels in the absence of a data set.

Table 5: Pruning Results without using the training dataset and no finetuning on CIFAR-100. RN is an abbreviation for ResNet; CP denotes if we choose to prune coupled channels; RF denotes the reduction in FLOPs; RP denotes the reduction in parameters; ITS denotes inference time speedup.

| Model Name | CP? | Acc-1(%) | RF | RP | ITS(CPU) | ITS(GPU) |
|---|---|---|---|---|---|---|
| Unpruned RN-50 | - | 78.85 | 1x | 1x | 1x | 1x |
| Random pruned RN-50 | No | 70.29 | 1.08x | 1.08x | 1.06x | 1.04x |
| L1-norm prunedLi et al. (2017) RN-50 | No | 70.24 | 1.16x | 1.02x | 1.17x | 1.08x |
| DFPC pruned RN-50 | No | 71.75 | **1.23**x | **1.20**x | **1.31**x | **1.11**x |
| Random pruned RN-50 | Yes | 69.50 | 1.07x | 1.07x | 1.02x | 1.02x |
| L1-norm prunedLi et al. (2017) RN-50 | Yes | 69.61 | 1.21x | 1.02x | 1.12x | **1.18**x |
| DFPC pruned RN-50 | Yes | 70.31 | **1.27**x | **1.22**x | **1.24**x | 1.16x |
| Unpruned RN-101 | - | 79.43 | 1x | 1x | 1x | 1x |
| Random pruned RN-101 | No | 71.66 | 1.11x | 1.10x | 1.07x | 1.05x |
| L1-norm prunedLi et al. (2017) RN-101 | No | 70.07 | 1.30x | 1.18x | 1.32x | 1.13x |
| DFPC pruned RN-101 | No | 70.01 | **1.71**x | **1.53**x | **1.54**x | **1.30**x |
| Random pruned RN-101 | Yes | 71.68 | 1.08x | 1.08x | 1.05x | 1.02x |
| L1-norm prunedLi et al. (2017) RN-101 | Yes | 71.59 | 1.25x | 1.12x | 1.20x | 1.16x |
| DFPC pruned RN-101 | Yes | 70.03 | **1.72**x | **1.53**x | **1.82**x | **1.34**x |
| Unpruned VGG-19 | - | 72.02 | 1x | 1x | 1x | 1x |
| Random pruned VGG-19 | - | 68.92 | 1.02x | 1.02x | 1.00x | 1.00x |
| L1-norm prunedLi et al. (2017) VGG-19 | - | 70.40 | 1.16x | 1.31x | 1.14x | 1.06x |
| DFPC pruned VGG-19 | - | 70.10 | **1.26**x | **1.50**x | **1.20**x | **1.11**x |

Table 6: Pruning Results without using the training dataset and no finetuning for the MobileNet-v2 architecture. MV2 is an abbreviation for MobileNet-v2; RF denotes the reduction in FLOPs; RP denotes the reduction in parameters; ITS denotes inference time speedup.

| Model Name | Dataset | Acc-1(%) | RP | RF | ITS(CPU) | ITS(GPU) |
|---|---|---|---|---|---|---|
| Unpruned MV2 | CIFAR-10 | 92.5 | 1x | 1x | 1x | 1x |
| L1-norm prunedLi et al. (2017) MV2 | CIFAR-10 | 90.36 | 3.92x | 3.60x | **3.62**x | **2.31**x |
| DFPC pruned MV2 | CIFAR-10 | 90.08 | **5.32**x | **3.74**x | 3.54x | 2.23x |
| Unpruned MV2 | CIFAR-100 | 72.78 | 1x | 1x | 1x | 1x |
| L1-norm prunedLi et al. (2017) MV2 | CIFAR-100 | 71.87 | 2.70x | 2.91x | 3.21x | 2.12x |
| DFPC pruned MV2 | CIFAR-100 | 69.87 | **3.61**x | **3.16**x | **3.25**x | **2.15**x |

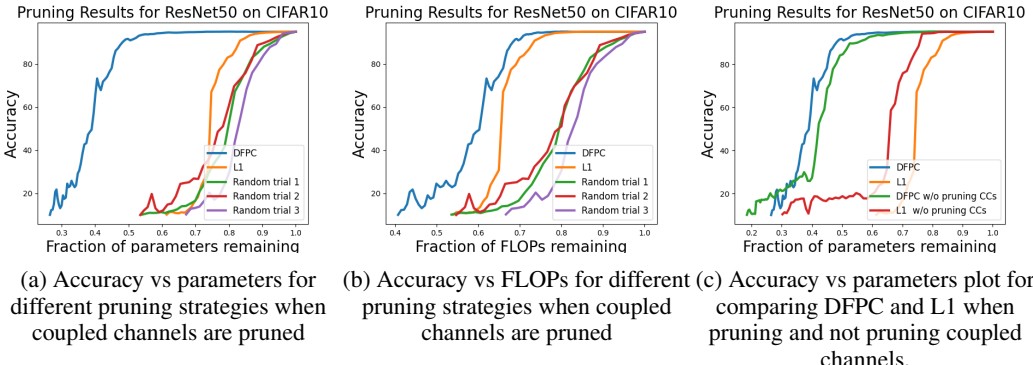

(a) Accuracy vs parameters for different pruning strategies when coupled channels are pruned

(b) Accuracy vs FLOPs for different pruning strategies when coupled channels are pruned

(c) Accuracy vs parameters plot for comparing DFPC and L1 when pruning and not pruning coupled channels.

Figure 7: Plots for pruning experiments on the ResNet-50 architecture trained on CIFAR-10 dataset

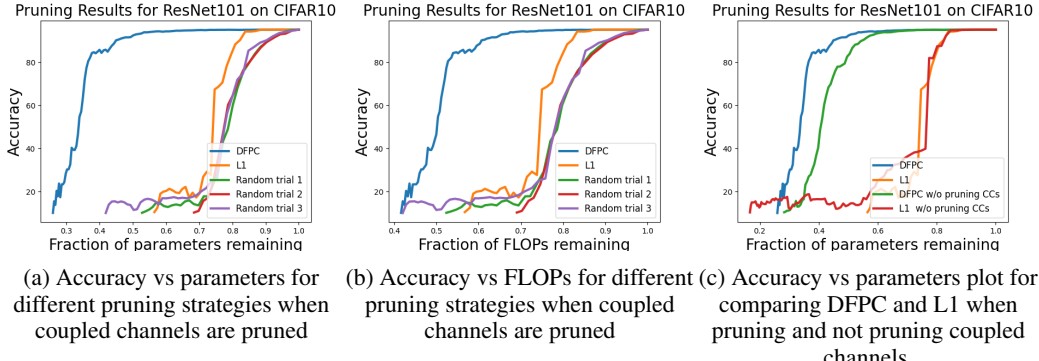

(a) Accuracy vs parameters for different pruning strategies when coupled channels are pruned

(b) Accuracy vs FLOPs for different pruning strategies when coupled channels are pruned

(c) Accuracy vs parameters plot for comparing DFPC and L1 when pruning and not pruning coupled channels.

Figure 8: Plots for pruning experiments on the ResNet-101 architecture trained on CIFAR-10 dataset

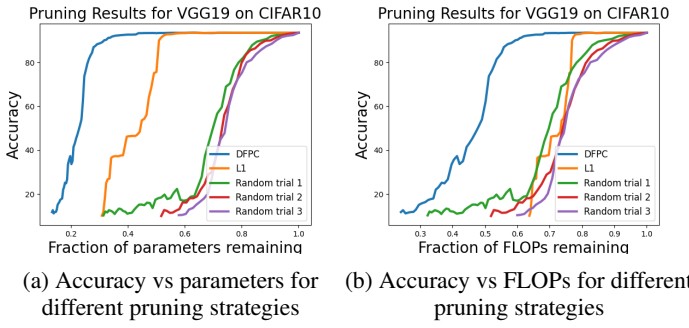

(a) Accuracy vs parameters for different pruning strategies

(b) Accuracy vs FLOPs for different pruning strategies

Figure 9: Plots for pruning experiments on the VGG19 architecture trained on CIFAR-10 dataset

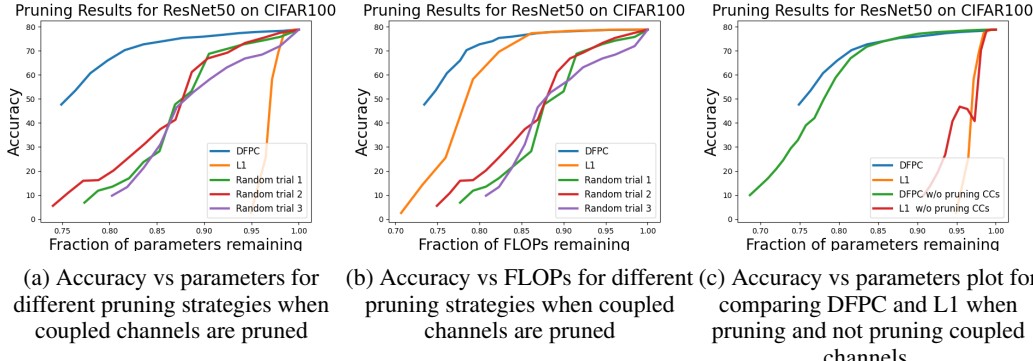

(a) Accuracy vs parameters for different pruning strategies when coupled channels are pruned

(b) Accuracy vs FLOPs for different pruning strategies when coupled channels are pruned

(c) Accuracy vs parameters plot for comparing DFPC and L1 when pruning and not pruning coupled channels.

Figure 10: Plots for pruning experiments on the ResNet-50 architecture trained on CIFAR-100 dataset

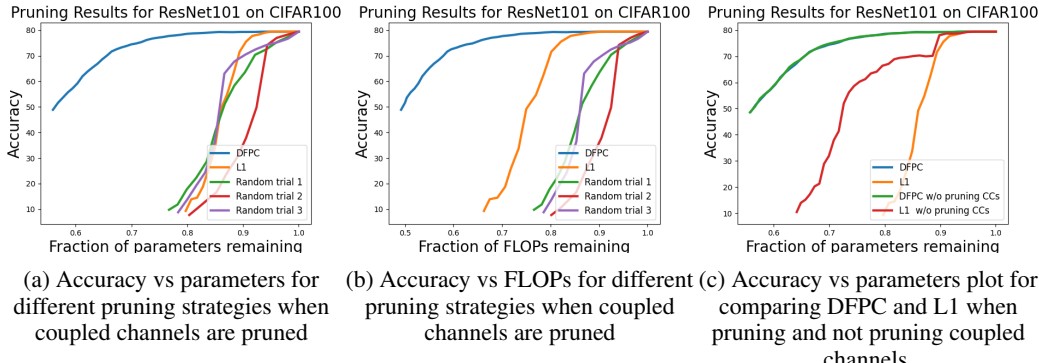

(a) Accuracy vs parameters for different pruning strategies when coupled channels are pruned

(b) Accuracy vs FLOPs for different pruning strategies when coupled channels are pruned

(c) Accuracy vs parameters plot for comparing DFPC and L1 when pruning and not pruning coupled channels.

Figure 11: Plots for pruning experiments on the ResNet-101 architecture trained on CIFAR-100 dataset

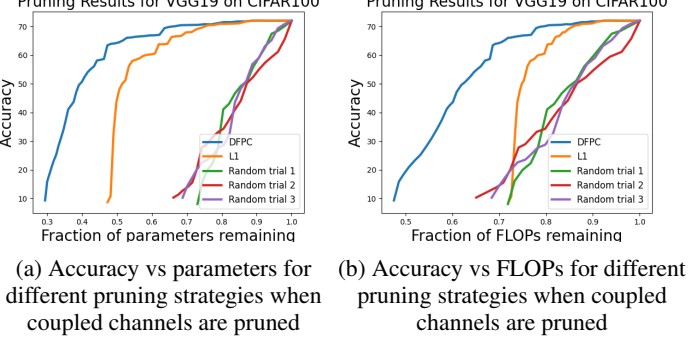

(a) Accuracy vs parameters for different pruning strategies when coupled channels are pruned

(b) Accuracy vs FLOPs for different pruning strategies when coupled channels are pruned

Figure 12: Plots for pruning experiments on the VGG19 architecture trained on CIFAR-100 dataset

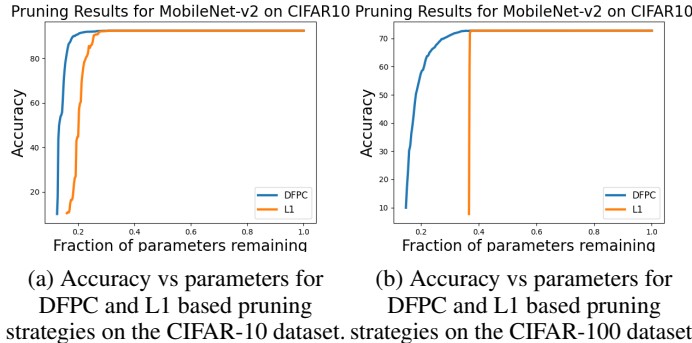

(a) Accuracy vs parameters for
DFPC and L1 based pruning
strategies on the CIFAR-10 dataset.

(b) Accuracy vs parameters for
DFPC and L1 based pruning
strategies on the CIFAR-100 dataset.

Figure 13: Plots for pruning experiments on the MobileNet-v2 architecture trained on CIFAR-10 and CIFAR-100 datasets.

### F.2 IMAGENET EXPERIMENTS

#### F.2.1 WITHOUT FINETUNING (DATA-FREE REGIME)

In this Section, we present the results of Pruning on the ImageNet dataset. We perform the following set of experiments. For ResNet-50, and ResNet-101 we measure accuracy vs sparsity(in terms of parameters) for the ImageNet dataset. These experiments are carried out for two grouped saliencies: DFPC and L1. Moreover, we performed these experiments two times. We prune both coupled and non-coupled channels in this first set of experiments. But in the second set of experiments, we only prune the non-coupled channels.

**Pruning Procedure**. Once we obtain grouped saliencies $Sal_\tau(k)$ for each channel of every DFC in a network, we compare these scores globally to select the channel to prune among all DFCs.

**Pretrained Models**. Pretrained models of ResNet-50 and ResNet-101 are obtained from the Torchvision library.

**Figures**. In figure 14, we plot the results of our pruning experiments to show how accuracy varies with parametric sparsities when we choose to prune coupled channels for the two strategies.

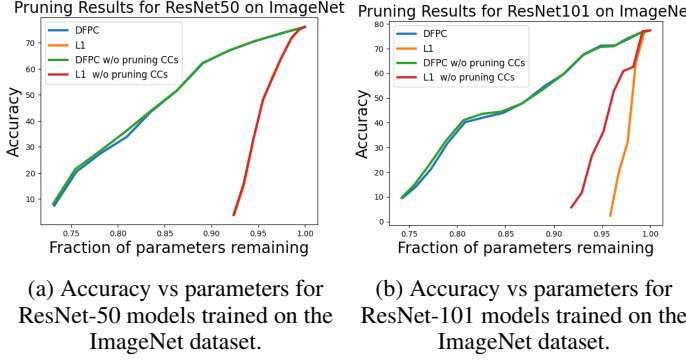

(a) Accuracy vs parameters for
ResNet-50 models trained on the
ImageNet dataset.

(b) Accuracy vs parameters for
ResNet-101 models trained on the
ImageNet dataset.

Figure 14: Plots for Accuracy vs parameters for ResNet-50 models trained on the ImageNet dataset.

**Discussion**. The accuracies drop quite quickly for models trained on the ImageNet dataset. However, we still find that DFPC obtained better sparsities than the L1 score for both cases when we pruned coupled channels and when we didn't. Moreover, we see that the trajectories of pruning are quite similar in terms of accuracy vs sparsity, irrespective of choosing to prune coupled channels. This could be attributed to the quick drop in accuracy of this experiment. Due to a quick drop in accuracies, we could not find a suitable accuracy level where we could report speedup fairly.

**Comparision with Yin et al. (2020)**. Yin et al. (2020) is a contemporary work in Data-Free pruning that synthesizes the dataset from a pre-trained model. Synthesis of such a dataset is computationally expensive. In this comparison, we compare the reduction in FLOPs vs the

accuracy drop of Yin et al. (2020) and DFPC. For a 1.02x FLOP reduction, the Accuracy of DFPC drops to 70.8%. However, Yin et al. (2020) maintain a 76.1% accuracy for a FLOP reduction of 1.3x.

### F.2.2 WITH FINETUNING (DATA-DRIVEN REGIME)

In this Section, we present the experimental of our pruning experiments on ResNet-50 trained on ImageNet dataset when we finetune the model as presented in Table 2 in Section 6 of the manuscript.

**Experimental Setup**. We use the pre-trained model of ResNet-50 available as a part of Torchvision for pruning. We prune 1% of the remaining channels in each pruning iteration followed by a finetuning of 3 epochs, each with step sizes of $10^{-3}, 10^{-4}, 10^{-5}$ per pruning iteration. The batch size was 256. After the pruning ends, we finally prune the network for 90 epochs with a batch size of 512. We use the SGD Optimizer with a momentum factor of 0.9 and weight decay of $1 \times 10^{-4}$ and Cosine Annealed step sizes with an initial learning rate of 0.1. Here, we normalize the saliency scores of each DFC during each pruning iteration.

**Pruned ResNet-50 architectures obtained**. In Figure 15, we present the pruned architectures of ResNet-50 obtained on pruning and finetuning when using DFPC. We see that all layers are actually being pruned and that layers within the sets in the following list are being pruned which were not pruned by most structured pruning works. Moreover notice that all members within a particular set have the same number of remaining channels.

- $\{conv1\}$

- $\{layer1.0.downsample.0,\ layer1.0.conv3,\ layer1.1.conv3,\ layer1.2.conv3\}$

- $\{layer2.0.downsample.0,\ layer2.0.conv3,\ layer2.1.conv3,\ layer2.2.conv3,\ layer2.3.conv3\}$

- $\{layer3.0.downsample.0,\ layer3.0.conv3,\ layer3.1.conv3,\ layer3.2.conv3,\ layer3.3.conv3,\ layer3.4.conv3,\ layer3.5.conv3\}$

- $\{layer4.0.downsample.0,\ layer4.0.conv3,\ layer4.1.conv3,\ layer4.2.conv3\}$

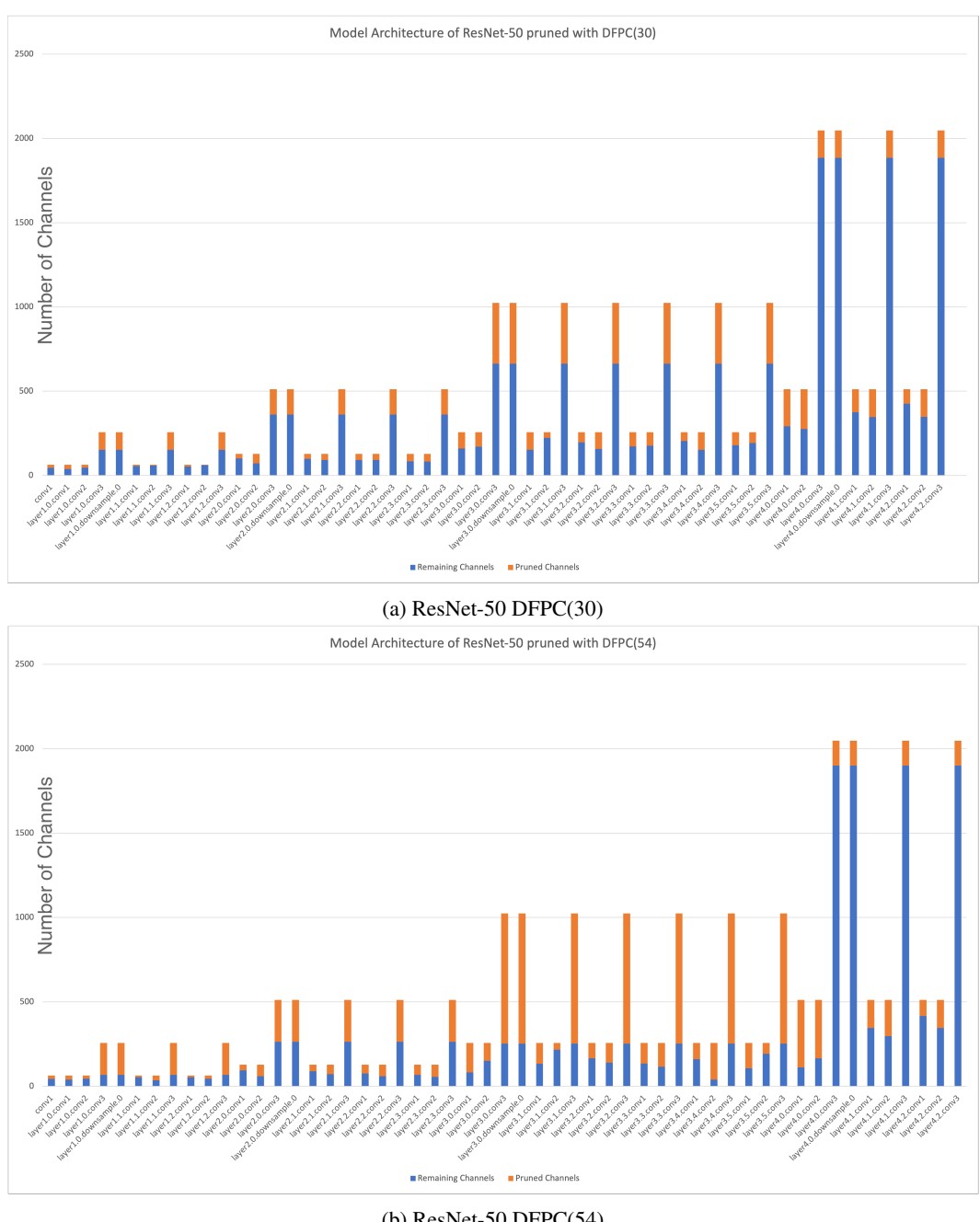

(a) ResNet-50 DFPC(30)

(b) ResNet-50 DFPC(54)

Figure 15: Visualization of ResNet-50 architecture pruned using DFPC.

