# OpenReview forum: "DFPC: Data flow driven pruning of coupled channels without data."
_ICLR.cc/2023/Conference — ICLR 2023 poster_

### Official Review · Reviewer_oXKf · 2022-10-24

**Confidence:** 3
**Correctness:** 3
**Technical Novelty And Significance:** 3
**Empirical Novelty And Significance:** 3
**Recommendation:** 6

**Clarity, Quality, Novelty And Reproducibility:**

The paper is relatively novel enough for its proposed DFPC. The content is presented clearly and organized reasonably. If the authors provide the source code for this paper, it will be easier for the reproducibility

**Strength And Weaknesses:**

Strengths
1. The paper targets the important problem of coupled channels for residual connections, which improves the performance of several classic networks.
2. The observation of disagreement for channel importance is interesting. Before this, metrics mainly used L1 or Taylor Approximate which doesn't take any channel dependency into account.
3. The overall writing is easy to follow.

Weaknesses
1. The authors claim that their method aims to prune channels like those fed into residual connections. However, they only conduct experiments on residual networks, and other popular networks such as mobilenets, densenets have no comparative experiments.
2. It's hard to read the text and understand Figure4.


**Summary Of The Paper:**

This paper proposes a method to prune deep neural networks. The motivation of this work is to consider DATA FLOW DRIVEN PRUNING OF COUPLED CHANNELS  (DFPC), which considers the coupled channels (CCs) in a data-free mode when conducting network pruning. The key part of this work is developing a method called DFCs that abstracts the end-to-end transformation of CCs. Then, a Maximum Score Disagreement mechanism is proposed for group saliency for DFCs. Experiments on VGG and ResNets show promising results across various image classification datasets, including Cifar10/100 and ImageNet.

**Summary Of The Review:**

Overall, I vote for borderline acceptance. Specifically, I liked the idea of investigating the disagreement mechanism among the coupled channels and proposing a new method for data-free pruning.

---

> ### Author Response · Authors · 2022-11-15
> **Response to Reviewer oXKf**
>
> Dear Reviewer oXKf,
>
> We are grateful for your valuable feedback on our paper and your interest in our methodology and score disagreement. We second that it is important to consider channel dependency when scoring coupled channels for pruning. We present our response to your comments below.
>
>  1. **If the authors provide the source code for this paper, it will be easier for the reproducibility.**
> 	 - For reproducibility, we have provided a google drive [link](https://drive.google.com/drive/folders/18eRYzWnB_6Qq0cYiSzvyOgicqn50g3-m?usp=sharing) to our code in the Appendix (page 12) of the submitted draft. This link may be missed in its position; thus, we move the link in the footnote on the first page of the revised manuscript. Moreover, upon acceptance, we plan to put up a public link to a GitHub repository for our code.
>  2. **The authors claim that their method aims to prune channels like those fed into residual connections. However, they only conduct experiments on residual networks, and other popular networks such as mobilenets, densenets have no comparative experiments.**
> 	 - Experiments on a different architecture will indeed strengthen our methodology. We perform additional pruning experiments on MobileNet-v2 and report them for the CIFAR-10 and CIFAR-100 datasets below. In our updated manuscript, we add these results to Appendix G.1 of the supplementary in Table 6 and corresponding plots in Figure 13. We observe similar trends for MobileNet-v2 as seen for other architectures and datasets.
>  3. **It's hard to read the text and understand Figure 4.**
> 	 - Thank you for pointing this out. We have updated this in our revised manuscript.
>
> Finally, we thank you for taking the time to review our paper. We have updated the manuscript with the said changes and would appreciate your confirmation that our response addresses your comments. We would be happy to have any further discussions.

---

> > ### Author Response · Authors · 2022-12-11
> > **Additional Response to Reviewer oXKf**
> >
> > Dear Reviewer oXKf,
> >
> > We wanted to express our gratitude for reviewing our submission. Your feedback was very helpful, and we have done our best to address it in our rebuttal. As the second phase of the rebuttal process is coming to a close, we would be grateful if you could acknowledge receipt of our rebuttal and let us know if it addresses your comments. We would also be happy to engage in further discussions if needed. Thank you again for your expertise and assistance.

---

### Official Review · Reviewer_6irU · 2022-10-24

**Confidence:** 2
**Clarity, Quality, Novelty And Reproducibility:** The article was well written, and the…
**Correctness:** 3
**Technical Novelty And Significance:** 3
**Empirical Novelty And Significance:** 3
**Recommendation:** 6

**Strength And Weaknesses:**

The algorithms and parameters were well outlined in this paper. In total, the contributions made in the paper were well set out.

More data visualisations could have been done to show the impact of this approach

**Summary Of The Paper:**

This paper focused on coupling the data flow to outputs and inputs, allowing for interconnections between outputs of
multiple layers. This led into the concept to prune CC without data to obtain aster latencies.

**Summary Of The Review:**

Overall, this paper adds value to the body of knowledge.

---

> ### Author Response · Authors · 2022-11-15
> **Response to Reviewer 6irU**
>
> Dear reviewer 6irU,
>
> We are grateful for your time to review our paper and that you find value in our work. We thank you for appreciating the presentation of our content.
> 1. **More data visualisations could have been done to show the impact of this approach.**
> 	- In the submitted draft, we already present visualizations of accuracy vs sparsity for different models and on different datasets in the Appendix for ablation. These plots are presented in Figures 7, 8, 9, 10, 11, 12, and 13(14 in the updated manuscript).
> 	- Additionally, as suggested by Reviewer oXKf, we present additional plots in Figure 13 of the updated manuscript for the MobileNet-v2 architecture.
> 	- In the updated manuscript, we present visualizations (Figure 15) of the DFPC pruned architecture of ResNet-50 for the ImageNet experiment with finetuning in Appendix G2.2 This visualization shows that layers with coupled channels are being pruned, and those that belong to the same DFC have the same number of output channels remaining.
>
> We have revised the manuscript with your kind suggestions and would be grateful if you could confirm that our response addresses your comments. We would be happy to be a part of any further discussions.

---

> > ### Author Response · Authors · 2022-12-11
> > **Additional Response to Reviewer 6irU**
> >
> > Dear Reviewer 6irU,
> >
> > Thank you for your valuable feedback on our submission. We have read your comments carefully and have addressed them in our rebuttal. As the second phase of the rebuttal process is ending soon, we would be grateful if you could acknowledge if our responses have addressed your comments. We would also be happy to engage in further discussions if needed. Thank you again for your time and consideration.

---

### Official Review · Reviewer_gfw2 · 2022-10-24

**Confidence:** 3
**Correctness:** 3
**Technical Novelty And Significance:** 3
**Empirical Novelty And Significance:** 3
**Recommendation:** 8

**Clarity, Quality, Novelty And Reproducibility:**

In general, the paper is clearly written and enough details are provided to reproduce the experiments.

**Strength And Weaknesses:**

Strength:
- The problem considered is an important one, and the proposed method is technically sound.
- Experiments conducted show that the proposed method achieves good empirical performance.
- The paper is well written in general.

Weaknesses:
- The proposed BGSC algorithm seems quite computationally expensive. While there is an analysis of the time complexity, it would be nice to also report the practical time taken to run the algorithm in practice. Does the proposed method take up significantly more time?
- I feel like the BGSC algorithm needs more explanation. While the overview gave a good insight into the algorithm, there are a lot of technical details in the algorithm that needs to be further discussed and elaborated.

**Summary Of The Paper:**

This paper tackles an important problem of neural network pruning. Specifically, the authors of the paper propose a novel method to prune coupled channels in neural networks. For instance, the layers with skip connections in the ResNet model are considered to be coupled channels. This is an under-considered problem, and most of the previously proposed pruning methods neglect these coupled channels. Nevertheless, coupled channels consist of a significant portion of many modern neural network architectures. in essence, the proposed approach involves traversing through all the paths between any pair of such couple channels, and aggregating the overall pruning scores for all of these individual paths. Then, an overall score can be determined and such couple channels can be effectively pruned. The authors of the paper conduct careful experiments to demonstrate the effectiveness of the proposed method.

**Summary Of The Review:**

All in all, I have some minor concerns regarding the computational cost of the proposed method. nevertheless, I think this is a good paper and think that it should be accepted.

---

> ### Author Response · Authors · 2022-11-15
> **Response to Reviewer gfw2**
>
>
> Dear Reviewer gfw2,
>
> We thank you for your valuable feedback and that you find strength in our problem statement and methodology. We agree that this is an under-considered problem. We respond to your comments below.
> 1. **The proposed BGSC algorithm seems quite computationally expensive. While there is an analysis of the time complexity, it would be nice to also report the practical time taken to run the algorithm in practice. Does the proposed method take up significantly more time?**
>    - The BGSC Algorithm presented as Algorithm 1 in the submitted draft of the paper is solely put forth for exposition purposes. This Algorithm admits an inherent parallelism which was exploited through multiprocessing/multithreading in our implementation to obtain faster saliency computation for the experiments conducted in the submitted draft (details presented in Appendix E.3.2). We report our execution time for the multithreading-based implementation below for our CPU hardware (specified in appendix A.1) when each method can create up to 10 threads.
>
>    - | Model             | Dataset        | Time Taken        |
>      | -- | -- | -- |
>      | ResNet-50              | ImageNet      | 38 minutes |
>      | ResNet-50              | CIFAR-10/100      | 12 minutes |
>      | MobileNet-v2              | CIFAR-10/100      | 68 seconds |
>
>    - We have added MobileNet-v2 experiments to the updated draft upon the suggestion of reviewer oXKf. We add Appendix E.3.3 in our manuscript to specify the saliency computation time.
>
> 2. **I feel like the BGSC algorithm needs more explanation. While the overview gave a good insight into the algorithm, there are a lot of technical details in the algorithm that needs to be further discussed and elaborated.**
>
>    - In the submitted version of our manuscript, we already present proof of correctness in Appendix D for the BGSC Algorithm. Moreover, in the submitted manuscript, Appendix E elaborates comprehensively on the technical details of how the BGSC algorithm is implemented for CNNs. In the updated manuscript, we present additional details that serve as a gist to the proof towards the end of Appendix D.
>
> We are grateful for your time to review our paper. We have updated the manuscript with the said changes and would appreciate you confirming that our response addresses your comments. We look forward to further discussions.

---

> > ### Author Response · Authors · 2022-12-11
> > **Additional Response to Reviewer gfw2**
> >
> > Dear Reviewer gfw2,
> >
> > Thank you so much for taking the time to review our submission. We sincerely appreciate your detailed feedback and appraisal of our work, and have carefully considered your comments in our rebuttal response. As the second phase of the rebuttal process is coming to an end, we would be grateful if you could acknowledge receipt of our responses and let us know if they address your concerns. We are eager to engage in any further discussions if needed.

---

### Author Response · Authors · 2022-11-15
**Revised manuscript uploaded.**

We thank all the reviewers for their helpful feedback and suggestions. We have responded to their comments individually. We have also uploaded an updated version of the manuscript incorporating their suggestions. The main changes in this version of the manuscript are:

1. Added results of additional data-free experiments on MobileNet-v2 for CIFAR-10/100 datasets.
2. Moved the code link from the Appendix to the footnote of the first page for visibility.
3. Added execution time of BGSC for different architectures on different datasets to Appendix E.3.2.
4. Added gist to proof of correctness of the BGSC algorithm in Appendix D
5. Updated captions of Figure 4 for better readability.
6. Added visualisations of pruned architectures of ResNet-50 in the data-driven regime to display the pruning of coupled channels.

We hope that these changes strengthen the state of our submission.

---

### Decision · Program_Chairs · 2023-01-20

**Decision:**

Accept: poster

**Justification For Why Not Higher Score:**

The proposed method seems to be meaningfully better, but it is not a dramatic improvement. It's also a nice problem.

**Justification For Why Not Lower Score:**

Reviewers all recommended acceptance. I am not super familiar with the literature on pruning (it's a total mystery to me why I keep getting assigned papers in this area) so it's possible it could be rejected. But I trust the reviewers here.

**Metareview: Summary, Strengths And Weaknesses:**

This paper focuses on structured pruning of neural networks, with a particular focus on pruning "coupled channels" - activations that are connected in the neural network through residual connections. The paper proposes a data-free approach (i.e. not requiring additional fine-tuning or access to data) to identifying which coupled channels should be pruned. The method outperforms prior pruning methods in terms of speed and does not sacrifice too much accuracy. Reviewers all felt the paper was strong enough to warrant publication, particularly after the rebuttal. The authors should update their figures to make them readable, since the text is much too small.

**Note From Pc:**

if the above contains the word "oral" or "spotlight" please see: "oral" presentation means -> notable-top-5% and "spotlight" means -> notable-top-25%. As stated in our emails, we are disassociating presentation type from AC recommendations